# The transcription factor RUNX2 drives the generation of human NK cells and promotes tissue residency

Sigrid Wahlen[1,2], Filip Matthijssens[2,3], Wouter Van Loocke[2,3], Sylvie Taveirne[1,2†], Laura Kiekens[1,2], Eva Persyn[1,2], Els Van Ammel[1,2], Zenzi De Vos[1,2], Stijn De Munter[1,2], Patrick Matthys[4], Filip Van Nieuwerburgh[5], Tom Taghon[1,2], Bart Vandekerckhove[1,2], Pieter Van Vlierberghe[2,3], Georges Leclercq[1,2]*

[1]Department of Diagnostic Sciences, Ghent University, Ghent, Belgium; [2]Cancer Research Institute Ghent, Ghent, Belgium; [3]Department of Biomolecular Medicine, Ghent University, Ghent, Belgium; [4]Laboratory of Immunobiology, Rega Institute for Medical Research, Department of Microbiology, Immunology and Transplantation, KU Leuven, Leuven, Belgium; [5]Department of Pharmaceutics, Ghent University, Ghent, Belgium

**Abstract** Natural killer (NK) cells are innate lymphocytes that eliminate virus-infected and cancer cells by cytotoxicity and cytokine secretion. In addition to circulating NK cells, distinct tissue-resident NK subsets have been identified in various organs. Although transcription factors regulating NK cell development and function have been extensively studied in mice, the role of RUNX2 in these processes has not been investigated, neither in mice nor in human. Here, by manipulating RUNX2 expression with either knockdown or overexpression in human haematopoietic stem cell-based NK cell differentiation cultures, combined with transcriptomic and ChIP-sequencing analyses, we established that RUNX2 drives the generation of NK cells, possibly through induction of IL-2Rβ expression in NK progenitor cells. Importantly, RUNX2 promotes tissue residency in human NK cells. Our findings have the potential to improve existing NK cell-based cancer therapies and can impact research fields beyond NK cell biology, since tissue-resident subsets have also been described in other lymphocyte subpopulations.

*For correspondence: georges.leclercq@ugent.be

Present address: †Orionis Biosciences, Ghent, Belgium

Competing interest: The authors declare that no competing interests exist.

## Editor's evaluation

This study provides new insights into the role of RUNX2 in the development and function of human NK cells. The impact of RUNX2 has been examined in in vitro models and in humanized mouse models to identify the pathways that might be impacted in vivo. The authors uncover a number of intriguing observations that impact the regulation of tissue-resident human NK cells identifying RUNX2 as another player in the regulation of NK cells.

## Introduction

Since natural killer (NK) cells were discovered 45 years ago, researchers have been studying fundamental processes that control NK cell development and function, in order to harness their potential in cancer immunotherapy. Nowadays, NK cell adoptive transfer is performed to treat various blood-borne and solid cancers (*Shimasaki et al., 2020*; *Wu et al., 2020*). The therapeutic benefit of using NK over T cells lies in their ability to recognise and kill tumour cells without causing graft-versus-host disease, which results from the unique way NK cell activation is regulated (*Shimasaki et al., 2020*).

By balancing signals of activating and inhibitory receptors, NK cells are able to eliminate targets that downregulated MHC class I molecules and thus have become invisible to the adaptive immune system (*Abel et al., 2018*). Aside from eliminating malignant cells, NK cells are also well known for their role in innate immunity against viral infections, so their use as an antiviral drug is certainly worth exploring (*Finton and Strong, 2012*).

Human NK cells arise from haematopoietic stem cells (HSC) in the bone marrow, which sequentially develop into stage 1, 2, and 3 progenitors in secondary lymphoid tissues. The mature stages of NK cell development, namely CD56$^{bright}$CD16$^-$ (stage 4) and CD56$^{dim}$CD16$^+$ (stage 5) NK cells, are defined by the acquisition of NK cell receptors including CD16, NKp44, NKp46, killer cell immunoglobulin-like receptors (KIR), NKG2A, and CD94. During these final stages, NK cells migrate to the circulation and periphery, where they can exert their function as natural born killers and immune response mediators. More specifically, effector molecules like perforin, granzymes, and death ligands are released upon NK cell activation and ensure apoptosis of targets. At the same time, cytokines, such as chemokines, IFN-γ, and TNF-α, are responsible for modulating other immune cells (*Abel et al., 2018*; *Scoville et al., 2017*; *Cichocki et al., 2019*).

Most of our knowledge on NK cells is based on studies that used peripheral blood (PB) as primary source. Although NK cells make up a considerable part of the circulating lymphocyte population, they also provide protection against pathogens at various sites in the body and can therefore be found in organs such as liver, bone marrow, secondary lymphoid tissues, thymus, lungs, uterus, skin, kidneys, gut, and adipose tissues (*Valero-Pacheco and Beaulieu, 2020*). Tissue-resident NK (trNK) cells are confined to these sites because they express adhesion molecules and chemokine receptors that facilitate tissue retention (*Melsen et al., 2016*; *Freud et al., 2017*; *Peng and Tian, 2017*). The unique phenotype of each resident subset is shaped by the microenvironment, which complicates trNK cell classification (*Vitale et al., 2020*). Nevertheless, a universal human trNK cell phenotype was established, namely CD69$^+$CXCR6$^+$EOMES$^{high}$TBET$^{low}$. In contrast, circulatory NK cells are EOMES$^{low}$T-BET$^{high}$ and express markers that enable them to remain in circulation or to recirculate, including CX3CR1, CCR7, CD62L, CD49e, S1PR1, and S1PR5 (*Melsen et al., 2016*; *Castriconi et al., 2018*; *Hashemi and Malarkannan, 2020*).

To ensure functionality and avoid pathogenesis, NK cell development needs to be tightly regulated. Data from extensive murine-based research revealed an intricate network of transcription factors acting as regulators in NK cell development, including Eomes, T-bet, Ets1, Runx proteins, etc. (*Geiger and Sun, 2016*). RUNX1, RUNX2, and RUNX3 are members of the highly conserved RUNX family of transcription factors and share the common Runt homology domain, which functions as both the DNA- and protein-binding domain. Together with CBFβ and other co-factors, these RUNX proteins form a stable transcriptional complex with a high affinity for consensus sequences (*Mevel et al., 2019*). In mice, there is ample evidence for the involvement of Runx1 and Runx3 in haematopoietic processes, like HSC maintenance and adaptive immune cell development, and knockout models of either Cbfβ or Runx3 have shed light on the role of Runx in NK cell and innate lymphoid cell 1 development and function (*Mevel et al., 2019*; *Ebihara et al., 2015*; *Seo and Taniuchi, 2020*). However, we still have a poor understanding of the exact role of RUNX factors in human NK cell development and function. Moreover, the role of RUNX2 in NK cell development has not been investigated before, neither in mice nor in humans.

In this study, we show that of the two principal isoforms, predominantly RUNX2-I is expressed during human NK cell development. In addition, using human HSC-based in vitro cultures for NK cell differentiation, we demonstrate that RUNX2 is an important transcriptional regulator of NK cell development, which promotes the acquisition of a tolerogenic trNK cell phenotype.

## Results

### RUNX2 drives generation of human mature NK cells

We first analysed protein expression of RUNX1, RUNX2, and RUNX3 using flow cytometry in human HSC and in stages 1–5 of NK cell development from bone marrow, tonsil, or PB of healthy donors (gating is shown in *Figure 1—figure supplement 1A*). Additionally, RUNX protein expression was measured in corresponding stages obtained from human cord blood (CB) HSC-based NK cell differentiation cultures. While RUNX1 expression was highest in early differentiation stages, the opposite was

true for RUNX3. In case of RUNX2, the expression reached its peak at stage 1 and stage 2 progenitors and declined thereafter. In line with previous studies (*Collins et al., 2019*; *Allan et al., 2017*; *Dogra et al., 2020*), RUNX2 expression was higher in CD56$^{bright}$CD16$^-$ (stage 4) compared to CD56$^{dim}$CD16$^+$ (stage 5) PB NK cells (*Figure 1A*).

As illustrated in *Figure 1B*, *RUNX2* expression is under control of two promoters, that give rise to two principal isoforms. Transcription starting from the proximal promoter (P2) results in the type-I (*RUNX2-I*) isoform, consisting of seven exons, whereas the type-II isoform (*RUNX2-II*) is a product of the distal promoter (P1), located upstream of P2, and therefore has one additional exon (*Mevel et al., 2019*; *Stock and Otto, 2005*). To identify which isoform is predominantly expressed in human NK cell development, we first investigated the epigenetic landscape of the *RUNX2* gene locus in human HSC and NK cells from publicly available ATAC-sequencing (ATAC-seq) and from H3K4me3 and H3K27ac ChIP-seq data (*Koues et al., 2016*), that respectively mark active promoters and poised or active promoters and enhancers. These data reveal that the proximal promoter region of the *RUNX2-I* is more accessible and active in HSCs and NK cells compared to the distal promoter of *RUNX2-II* (*Figure 1C*), suggesting that the former isoform is more likely to be expressed in NK cell development. This is confirmed by qPCR analysis with isoform-specific primers (*Figure 1—figure supplement 1B*) of in vitro generated human NK cell developmental stages and of human PB NK cells (ex vivo; *Figure 1D*).

In order to identify the role of RUNX2 in human NK cell development, we either induced knockdown using a RUNX2-specific shRNA that targets all isoforms, or we ectopically overexpressed RUNX2-I in CB-derived HSC by viral transduction. A scrambled *shRNA*-containing vector and empty *IRES/eGFP* vector were used as corresponding negative controls. Transduced HSCs were sorted and cultured in NK cell differentiation conditions. qPCR and/or flow cytometric analysis confirmed that knockdown and overexpression significantly reduced and increased RUNX2 expression, respectively (*Figure 1—figure supplement 1C, D*). Although RUNX2 knockdown did not affect HSC to stage 2 cell differentiation, it significantly reduced stage 3 progenitor and NK cell numbers (*Figure 1E*, top graphs). Inversely, overexpression of RUNX2-I led to severe reduction of HSC, stage 1 and stage 2 cell numbers, and accelerated NK cell generation, as NK cell numbers were significantly increased compared to the control at day 7 and day 14 of culture (*Figure 1E*, bottom graphs). Furthermore, RUNX2-I overexpression promoted the differentiation of CD16$^+$ stage 5 NK cells from 14 days of culture onwards (*Figure 1E*, bottom graphs, *Figure 1—figure supplement 1E*).

Altogether, these results demonstrate that RUNX2-I is primarily expressed throughout human NK cell differentiation. Whereas RUNX2 knockdown inhibits NK development, overexpression accelerates the generation of mature NK cells.

## RUNX2 controls human NK development possibly by regulating IL-2Rβ expression

To uncover the underlying mechanism of RUNX2, we first examined whether RUNX2 knockdown affected survival and/or proliferation of NK cells. Interestingly, apoptosis was unaffected, whereas proliferation was even increased compared to control NK cells (*Figure 2—figure supplement 1A-C*). Next, we investigated NK cell development in more detail. NK cells strongly depend on IL-15 signalling during development, which is why NK lineage commitment is achieved when stage 3 progenitors express IL-2Rβ, also known as a subunit of the IL-15 receptor (*Geiger and Sun, 2016*; *Wang and Malarkannan, 2020*). Here, we used a high-affinity IL-2Rβ antibody and stained cells after overnight IL-15-depleted culture to increase antibody binding. The results show that RUNX2-I overexpression significantly increased the frequency of IL-2Rβ$^+$ NK-committed progenitors after day 7 of culture (*Figure 2A*). However, no significant difference was detected after 14 days of overexpression culture nor after 7 or 14 days of RUNX2 knockdown cultures. The premature rise of NK-committed progenitors in overexpression cultures might be caused by RUNX2 either directly targeting IL-2Rβ expression, or by affecting the generation of NK-committed progenitors independent of IL-2Rβ. Interestingly, the RUNX2-specific ChIP-seq analysis of human PB NK cells supports the former hypothesis by demonstrating that *IL2RB* is a direct target gene of RUNX2 (*Figure 2B*, *Supplementary file 1*). Additional evidence for the direct regulation of IL-2Rβ expression was gathered by overexpressing RUNX2-I in ALL-SIL cells that are RUNX2$^{low}$IL-2Rβ$^{low}$ and by silencing RUNX2 in YTS cells that are RUNX2$^{high}$IL-2Rβ$^{high}$. Overexpression in ALL-SIL cells resulted in increased IL-2Rβ expression, while knockdown in YTS did not provoke a significant change (*Figure 2C*). Together, these findings suggest that RUNX2

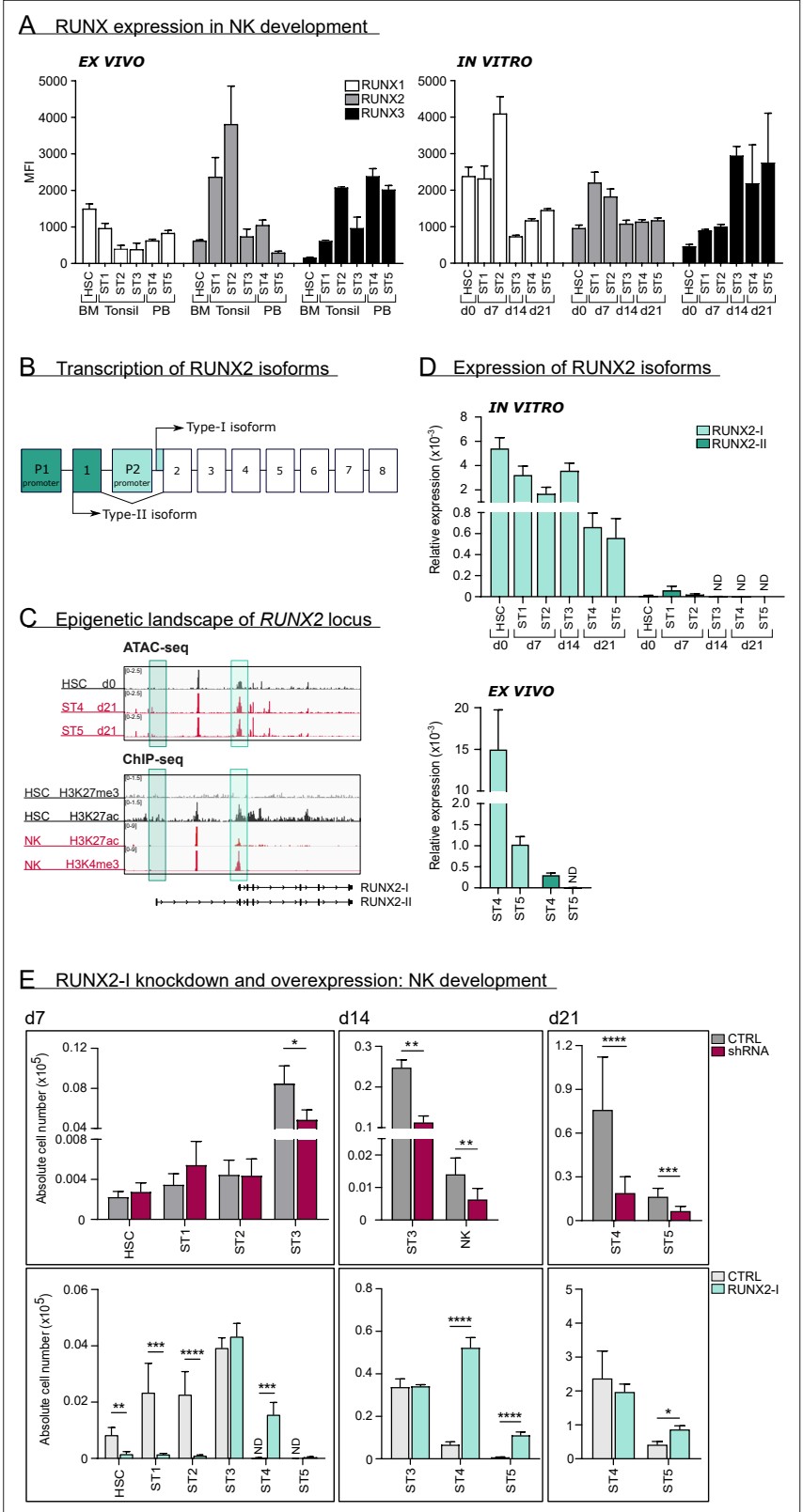

**Figure 1.** RUNX2-I is predominantly expressed and plays an important role in natural killer (NK) cell development. (**A**) RUNX1, RUNX2, and RUNX3 expression were evaluated in ex vivo and in vitro NK cell developmental stages with flow cytometry and presented as mean fluorescence intensity (MFI). For ex vivo data, haematopoietic stem cells (HSC; CD34$^+$CD45RA$^-$) originated from bone marrow, stage 1 (CD34$^+$CD45RA$^+$CD117$^-$), stage 2

*Figure 1 continued on next page*

*Figure 1 continued*

(CD34$^+$CD45RA$^-$CD117$^+$), and stage 3 (CD34$^-$CD117$^+$CD94$^-$HLA-DR$^-$NKp44$^-$) progenitors from tonsil, and stage 4 (CD56$^+$CD94$^+$CD16$^-$) and stage 5 (CD56$^+$CD94$^+$CD16$^+$) NK cells from peripheral blood (mean ± SEM; n=2–4). In vitro expression levels were determined in equivalent stages from cord blood (CB) HSC-based NK cell differentiation cultures at indicated time points (mean ± SEM; n=6). (**B**) Schematic overview of the transcriptional regulation of the RUNX2 principal isoforms. The *type II isoform* (*RUNX2-II*, eight exons) is transcribed from distal promoter *P1*, while the *type I isoform* (*RUNX2-I*, seven exons) is regulated by proximal promoter *P2*. (**C**) Genome browser tracks of *RUNX2* locus obtained from publicly available ATAC-seq data from HSC (d0), stages 4 and 5 NK cells (d21) of in vitro differentiation cultures (top panel) and histone ChIP-seq data from CB HSC (H3K27me3 and H2K27ac) and PB NK cells (H3K27ac and H3K4me3). The highlighted regions emphasise the promoter regions of the two *RUNX2* isoforms. (**D**) Relative expression of the *RUNX2* isoforms was measured in the indicated in vitro NK cell developmental stages and in ex vivo stages 4 and 5 PB NK cells using quantitative PCR (mean ± SEM; n=2–3). (**E**) CB-derived CD34$^+$ HSC were transduced with either a lentiviral vector containing a *RUNX2-specific shRNA* or a retroviral vector with the *RUNX2-I isoform* cDNA. A lentiviral vector containing *scrambled shRNA* and an empty retroviral vector were negative controls. Transduced eGFP$^+$ HSC (Lin$^-$CD34$^+$CD45RA$^-$) were sorted and cultured in NK cell-specific differentiation conditions. Absolute cell numbers of indicated NK cell developmental stages in RUNX2(-I) knockdown and overexpression differentiation cultures were determined using flow cytometry at the indicated time points (mean ± SEM; n=4–12). Statistical significance was determined using the paired Student's t-test. *, **, ***, and **** represent statistical significance compared to control-transduced cultures with p<0.05, p<0.01, p<0.001, and p<0.0001, respectively. ND, not detectable.

The online version of this article includes the following figure supplement(s) for figure 1:

**Figure supplement 1.** Gating strategy and expression of RUNX2 in knockdown and overexpression differentiation cultures.

probably promotes NK cell differentiation by direct induction of IL-2Rβ expression, hereby supporting NK lineage commitment.

## RUNX2 promotes the acquisition of a tissue-resident phenotype in human NK cells

To explore the RUNX2-mediated molecular mechanism in more depth, the RUNX2-specific ChIP-seq results and the results of RNA-sequencing of sorted RUNX2(-I) knockdown or overexpressing NK cells were analysed in parallel (*Supplementary files 1–4*). In the RUNX2 ChIP-seq, a total of 12,121 filtered peaks were identified of which the majority was located in the promoter region of target genes (*Figure 3A*, *Supplementary file 1*). The *RUNX* motif was most highly enriched in the motif analysis, which further validated the experiment (*Figure 3B*). In addition, *ETS1* and albeit with lower significance, *TBX21* motifs were found with high frequency (*Figure 3B*), suggesting that RUNX2 is able to control gene expression by forming a transcriptional complex with these factors. Overlapping the ChIP-seq and RNA-seq results reveals that most of the differentially expressed genes were direct transcriptional targets of RUNX2 (*Figure 3C*, *Supplementary file 4*), further highlighting its importance as a transcriptional regulator.

Upon closer inspection of the transcriptome analysis, a considerable number of genes, associated with NK cell tissue residency and homing were differentially expressed (*Figure 3D*, *Supplementary files 2 and 3*). trNK cells express specific proteins that enable them to be retained in organs, while proteins that allow NK to (re)circulate are downregulated. Several studies have already used RNA-seq analysis to identify NK cell resident- and circulatory-specific markers in the liver (*Cuff et al., 2016*) and bone marrow (*Melsen et al., 2018*). We used these gene sets in a Gene Set Enrichment Analysis (GSEA) to confirm the key role of RUNX2 in NK cell tissue residency. As indicated in *Figure 3E* (top plots; *Supplementary file 5*), tissue-resident genes of both liver and bone marrow were highly enriched in RUNX2-I-overexpressing NK cells, while circulatory NK-specific genes, here presented as trNK-downregulated genes, were mainly expressed in control NK cells. Strikingly, the opposite was observed in knockdown cultures (*Figure 3E*, bottom plots, *Supplementary file 5*). Additionally, the public RNA-seq datasets show that *RUNX2* transcript levels of ex vivo liver as well as bone marrow trNK are increased compared to their circulatory counterparts (fold changes are 25 and 13, respectively; *Cuff et al., 2016*; *Melsen et al., 2018*). This further supports our hypothesis that RUNX2 plays an important role in the development of human trNK cells. To confirm these results, we analysed homing factors by flow cytometry. While knockdown led to decreased expression of the tissue

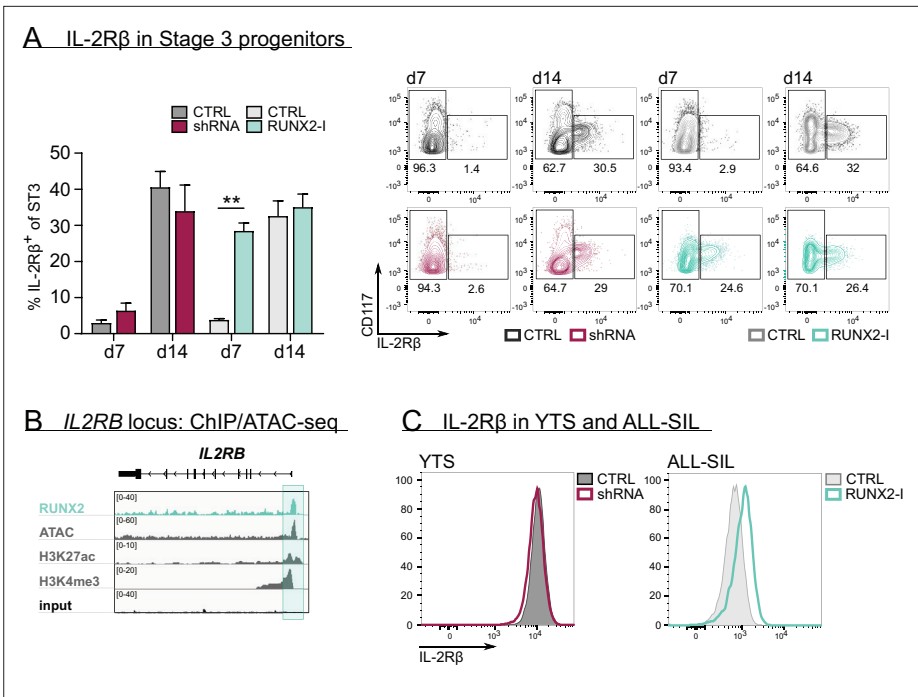

**Figure 2.** RUNX2 controls human natural killer (NK) development by directly regulating expression of IL-2Rβ. (**A**) The frequency of IL-2Rβ⁺ cells of stage 3 progenitors was determined by flow cytometry at day 7 and day 14 of RUNX2(-I) knockdown and overexpression cultures (mean ± SEM; n=3–4). The dot plots show representative CD117 versus IL-2Rβ stainings of gated stage 3 progenitors from the indicated cultures. (**B**) Genome browser tracks of the *IL2RB* locus of RUNX2 ChIP-seq of sorted human PB NK cells and of histone (H3K27ac and H3K4me3) ChIP-seq and ATAC-seq of PB NK cells. The significant RUNX2 ChIP peaks are marked in green. (**C**) RUNX2 knockdown and overexpression vectors were transduced in YTS and ALL-SIL cell lines, respectively. At 4 days after transduction, expression of IL-2Rβ was examined with flow cytometry. Statistical significance is determined using the paired Student's t-test. ** represents statistical significance compared to the control-transduced cultures with p<0.01.

The online version of this article includes the following figure supplement(s) for figure 2:

**Figure supplement 1.** RUNX2 plays no role in apoptosis and exerts a negative effect on proliferation of natural killer (NK) cells.

residency-specific markers CD69 and CD49a, circulatory markers such as CX3CR1, CCR7, CD62L, and S1PR1 were increased. On the contrary, forced expression of RUNX2-I generated NK cells with higher expression levels of CD69, CD49a, and CXCR4, and lower expression of CD49e and the above-mentioned circulation-specific markers, except for S1PR1 (*Figure 4A*). In addition, EOMES and T-BET expression can be used to distinguish resident (EOMES^highT-BET^low) and circulatory (EOMES^lowT-BET^high) NK cells in human (*Hashemi and Malarkannan, 2020*; *Collins et al., 2017*). Even though unaltered by knockdown, the frequency of EOMES^highT-BET^low and EOMES^lowT-BET^high NK cells in overexpression cultures was increased and decreased, respectively (*Figure 4B*). We also performed flow cytometric analysis of NK cell receptors, reported to be differentially expressed on trNK cells. Whereas RUNX2 knockdown only resulted in the reduced expression of NKp44, overexpression increased the frequency of NKG2C⁺ NK cells and the expression of NKp46 and NKG2A/CD94 but decreased the frequency of KIR⁺ NK cells (*Figure 4C*). Most genes encoding homing markers and NK cell receptors were also direct RUNX2 targets, as presented by ChIP-seq and ATAC-seq analysis in *Figure 4—figure supplement 1A and B*. In summary, these findings highlight a key role for RUNX2 in the acquisition of a tissue-resident phenotype in differentiating NK cells.

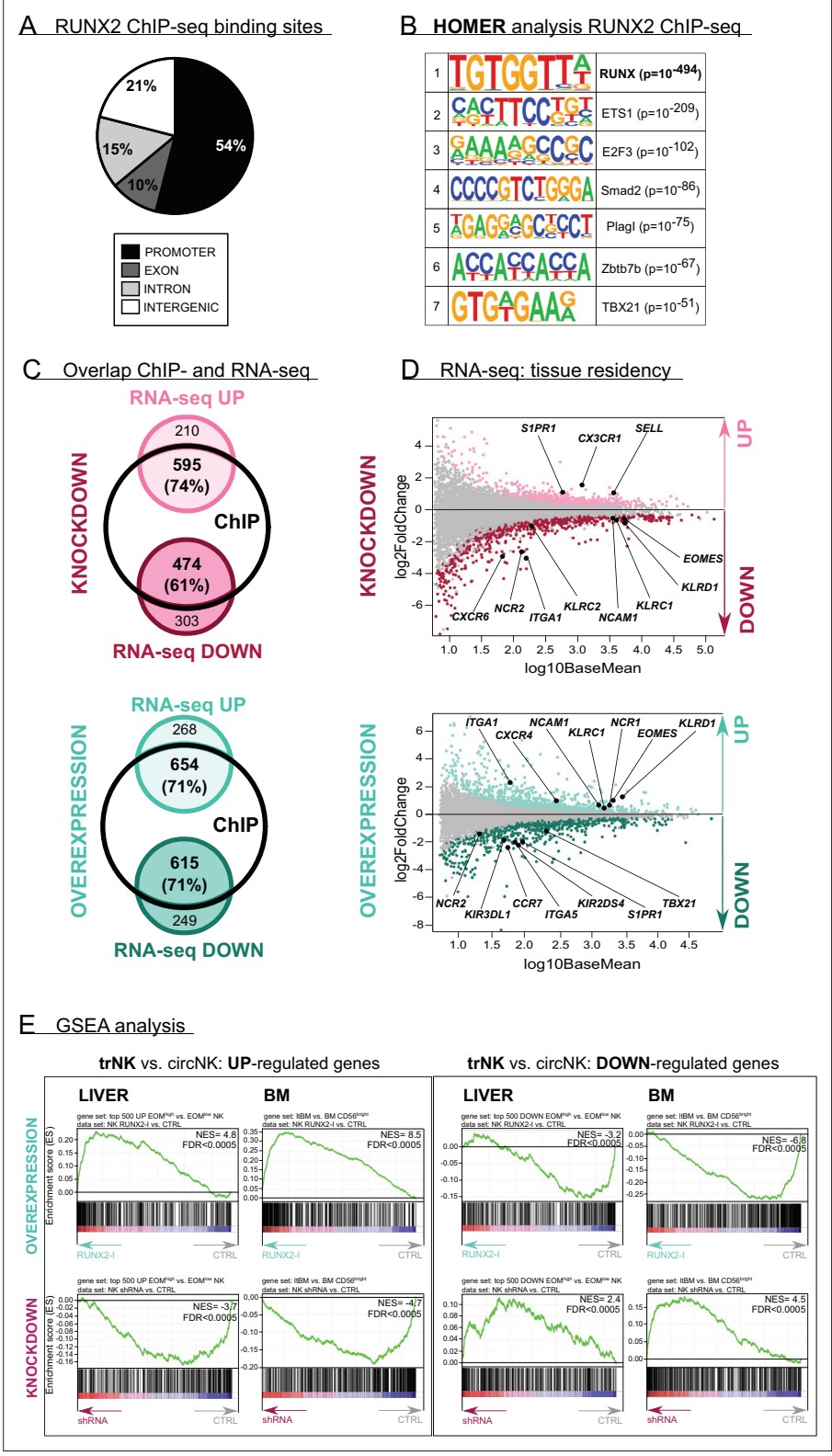

**Figure 3.** RUNX2 regulates a tissue residency transcriptional program. (**A–B**) RUNX2 ChIP-seq analysis was performed on sorted human PB natural killer (NK) cells. (**A**) Locations of RUNX2 ChIP peaks relative to genomic annotations. (**B**) The top 7 motifs obtained from the HOMER motif enrichment analysis of the RUNX2 ChIP-seq are depicted. (**C–E**) NK cells from RUNX2(-I) knockdown and overexpression cultures were sorted, and the transcriptome was analysed using RNA-sequencing (n=4–5). (**C**) The Venn diagrams show the overlap between

*Figure 3 continued on next page*

*Figure 3 continued*

ChIP-seq and the indicated RNA-seq analysis. The majority of the significantly up- ('UP') or downregulated ('DOWN') genes in both the knockdown and overexpression cultures were directly targeted by RUNX2. (**D**) MA plots displaying down- and upregulated genes in NK cells from RUNX2(-I) knockdown (top panel) and overexpression cultures (bottom panel). Tissue residency-associated genes are depicted. (**E**) Gene Set Enrichment Analysis (GSEA). The gene sets were obtained from studies comparing tissue-resident (trNK) and recirculating (circNK) NK cells in the liver (top 500 up- and downregulated genes; *Cuff et al., 2016*) or bone marrow (*Melsen et al., 2018*). Up- and downregulated genes in tissue-resident versus recirculating NK cell subsets are presented in the left and right box, respectively. The datasets were obtained by RNA-seq analysis of NK cells from RUNX2(-I) overexpression (top row) and knockdown cultures (bottom row).

## RUNX2 plays no role in cytotoxicity but inhibits cytokine and effector molecule production

The main functions of NK cells are killing malignant or virus-infected cells and shaping the local immune response by production of cytokines like IFN-γ and TNF-α (*Abel et al., 2018*. *Bruno et al., 2014*). Chromium release assays showed that killing by sorted NK cells from knockdown and overexpression cultures was not affected, despite RUNX2-silenced NK cells displaying decreased degranulation when co-incubated with K562 cells (*Figure 5A–B*). Whereas knockdown significantly increased granzyme B and perforin expression, overexpression resulted in a reduction or a negative trend in the expression of these effector molecules (*Figure 5C*).

To assess their cytokine response, cultured NK cells were stimulated with either phorbol myristate acetate (PMA)/ionomycin, or with IL-12/IL-18 in the presence or absence of IL-15. The production and/or secretion of IFN-γ and TNF-α was significantly increased or exhibited a positive trend in knockdown cultures. Inversely, overexpression resulted in the exact opposite, as shown in *Figure 5D and E*. These results imply that RUNX2 has a negative impact on the expression of cytotoxic effector molecules and cytokines. Additionally, the RUNX2-specific ChIP-seq analysis revealed that TNF-α and effector molecules were directly regulated by RUNX2 (*Figure 5—figure supplement 1*).

## In vivo evidence for the role of RUNX2 in human NK cell tissue residency

To determine whether RUNX2 plays a role in human NK cell development and tissue residency in vivo, we generated a humanised mouse model by intravenously injecting equal numbers of bulk trans-duced UCB-derived haematopoietic progenitor cells (HPC) into lethally irradiated *NSG-huIL-15* mice (*Figure 6A* and *Figure 6—figure supplement 1A*). The HPC were transduced with either control or *RUNX2 shRNA* lentivirus. After 6–7 weeks, we analysed the absolute numbers of NK cells in the lungs, liver, spleen, bone marrow, and intestinal lamina propria (LPL). NK cell numbers were drastically reduced in examined organs mice injected with RUNX2-silenced HPC compared to those that were injected with control HPC (*Figure 6B*). These findings show that RUNX2 is also required for human NK cell development in vivo. Next, the frequency of tissue-resident (CD69+CD49e−) and circulating (CD69−CD49e+) NK cells was examined. As shown in *Figure 6C*, the frequency of trNK cells is significantly reduced in the bone marrow and LPL fraction, while the percentage of circNK cells is increased (*Figure 6—figure supplement 1B*). This shows that in these organs, RUNX2 is involved in human NK cell tissue residency.

## Discussion

Human NK cell development has been subject to extensive research that characterised different consecutive differentiation stages in great detail (*Abel et al., 2018*; *Scoville et al., 2017*). However, the transcription factors that govern the transitions of these stages remain to be elucidated. Here, we identify an important role for RUNX2-mediated transcriptional regulation in human NK cell development and function. The primary isoform of RUNX2 expressed during human NK cell development is RUNX2-I, which is possibly required for NK lineage commitment via regulation of IL-2Rβ expression and drives the differentiation of NK cells towards a tolerogenic tissue-resident phenotype.

First, we examined RUNX protein expression in both ex vivo and in vitro NK cell development. While RUNX1 and RUNX3 display opposing expression profiles, RUNX2 is highly expressed in stage

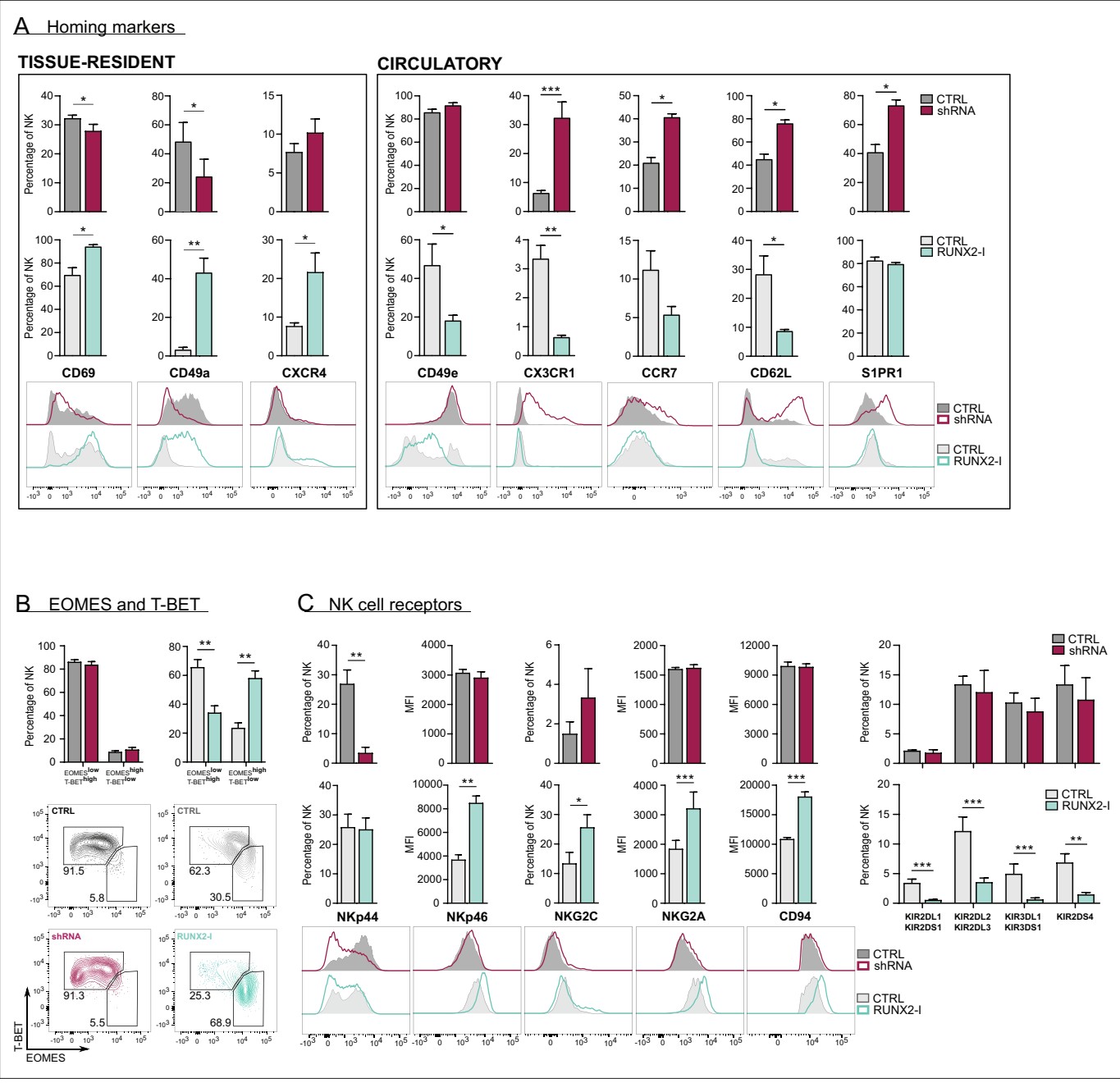

**Figure 4.** RUNX2 promotes a tissue-resident phenotype in human natural killer (NK) cells. (**A**) Expression of tissue-resident (CD69, CD49a, CXCR4) and circulation-specific factors (CD49e, CX3CR1, CCR7, CD62L, S1PR1) in NK cells of RUNX2(-I) knockdown and overexpression cultures, was checked with flow cytometry (mean ± SEM; n=4). Histograms display expression of markers in representative donors. (**B**) Percentage of NK cells with a circulatory (EOMES^lowT-BET^high) or tissue-resident (EOMES^highT-BET^low) phenotype, determined by flow cytometry (mean ± SEM; n=4). Dot plots represent typical samples. (**C**) The expression of NK cell receptors NKp44, NKp46, NKG2C, NKG2A, CD94, KIR2DL1, KIR2DS1, KIR2DL2, KIR2DL3, KIR3DL1, KIR3DS1, and KIR2DS4 in gated NK cells from RUNX2(-I) knockdown and overexpression cultures was measured with flow cytometry (mean ± SEM; n=3–9). Statistical significance was determined using the paired Student's t-test. *, **, and *** represent statistical significance compared to control-transduced cultures with p<0.05, p<0.01, and p<0.001, respectively.

The online version of this article includes the following figure supplement(s) for figure 4:

**Figure supplement 1.** Genome browser tracks of homing molecules, transcription factors, and natural killer (NK) cell receptors associated with tissue residency.

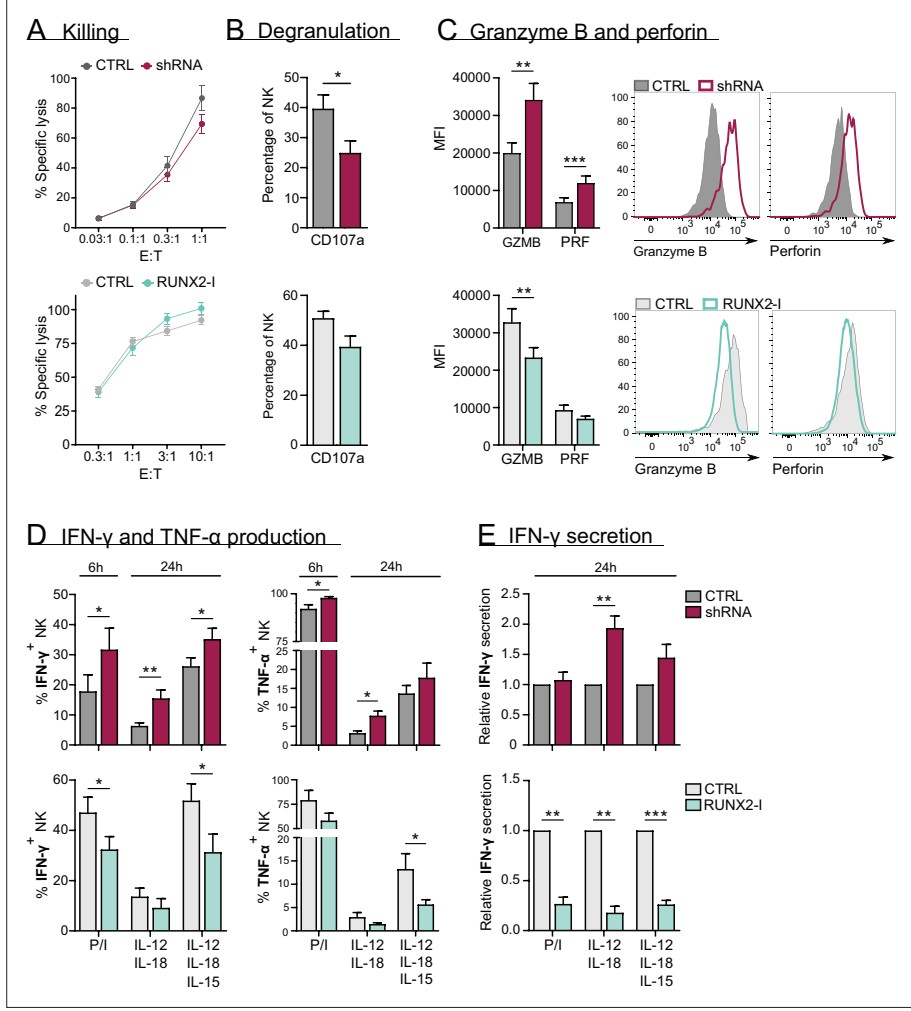

**Figure 5.** RUNX2 inhibits cytokine and cytotoxic effector molecule production, but has no impact on natural killer (NK) cell killing. (**A–C**) Different functional aspects of NK cells of RUNX2(-I) knockdown and overexpression cultures were examined. (**A**) Sorted NK cells were incubated with K562 target cells in a (***Bruno et al., 2014***) chromium release killing assay at the indicated effector:target ratios for 4 hr, and the percentage of specific lysis was determined (mean ± SEM; n=6–7). (**B**) Percentage CD107a$^+$ cells of NK cells as detected by flow cytometry after 2-hr co-culture with K562 cells (mean ± SEM; n=6–8). (**C**) Expression of cytotoxic effector molecules granzyme B (GZMB) and perforin (PRF). The mean fluorescence intensity was determined using flow cytometry (mean ± SEM; n=10). Histograms present expression of markers in representative donors. (**D**) Cells were stimulated in bulk with either phorbol myristate acetate (PMA)/ionomycin (6 hr), IL-12/IL-18, or IL-12/IL18/IL-15 (24 hr). IFN-γ and TNF-α production were analysed with flow cytometry (mean ± SEM; n=4–11). (**E**) Sorted NK cells were stimulated for 24 hr with either PMA/ionomycin, IL-12/IL-18, or IL-12/IL-18/IL-15. The supernatant was collected and the secretion of IFN-γ was analysed with ELISA (mean ± SEM; n=6). Statistical significance was determined using the paired Student's t-test. *, **, and *** represent statistical significance compared to control-transduced cultures with p<0.05, p<0.01, and p<0.001, respectively.

The online version of this article includes the following figure supplement(s) for figure 5:

**Figure supplement 1.** Genome browser tracks of effector molecule and cytokines.

1 and stage 2 progenitors, after which expression gradually wanes. Consistent with other studies (***Collins et al., 2019***; ***Allan et al., 2017***; ***Dogra et al., 2020***), RUNX2 levels are higher in stage 4 compared to stage 5 PB NK cells, which might be indicative for a significant role of RUNX2 in this subset. The expression of each RUNX protein is regulated by a proximal and distal promoter, that give rise to type-I and type-II isoforms, respectively (***Mevel et al., 2019***; ***Stock and Otto, 2005***). The functional differences of these isoforms are currently under intensive investigation. In case of Runx1, the type-II isoform seems to be important during early HSC formation, while the Runx1-I isoform is active

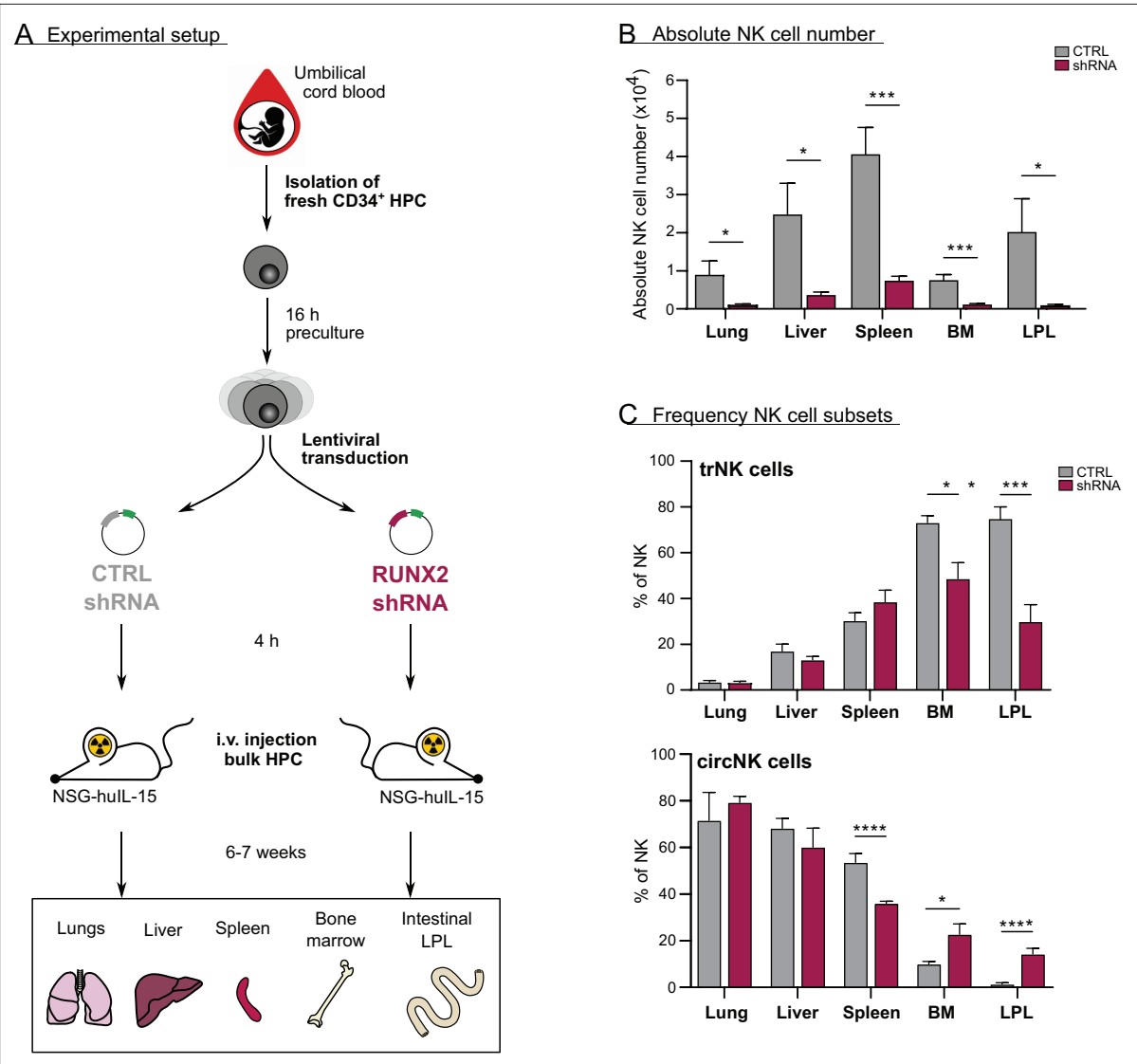

**Figure 6.** In vivo evidence for the role of RUNX2 in human natural killer (NK) cells tissue residency. (**A–C**) CD34+ HPC were isolated from fresh cord blood (CB) and cultured in preculture medium for 16 hr before lentiviral transduction with either control or *RUNX2 shRNA* virus. Approximately 4 hr later, the control- or *RUNX2 shRNA*-transduced HPC were intravenously injected in *NSG-huIL-15* mice, which were lethally irradiated. After 6–7 weeks, the presence of eGFP+ human NK cells (CD45+CD56+CD94+) (**B**) and the frequency of tissue-resident (CD69+CD49e−) and (**C**) circulating (CD69−CD49e+) NK cells were determined using flow cytometry (mean ± SEM; n=7). Statistical significance was determined using the unpaired Student's t-test. *, **, ***, and **** represent statistical significance compared to control mice with p<0.05, p<0.01, p<0.001, and p<0.0001, respectively. BM, bone marrow; HPC, haematopoietic progenitor cells; i.v., intravenous; LPL, lamina propria lymphocytes.

The online version of this article includes the following figure supplement(s) for figure 6:

**Figure supplement 1.** Control versus *RUNX2 shRNA* humanised mouse model.

later in haematopoietic cell differentiation (*Mevel et al., 2019*). Moreover, the expression of Runx3-I is dominant in all stages of murine NK cell development, although Runx3-II is still present in mature NK cells (*Ebihara et al., 2015*). Data concerning the expression of the Runx2 isoforms in NK cells is lacking for both mice and human. However, based on studies evaluating the expression of RUNX2 in human non-haematopoietic cells, it is suggested that RUNX2-II is bone-specific, whereas RUNX2-I is more broadly expressed in both osseous and non-osseous cell types (*Cuellar et al., 2020*; *Ogawa et al., 2000*). Here, we show, for the first time, that throughout human in vitro NK cell development and in ex vivo PB NK cells, mainly the *RUNX2-I isoform* is expressed. Strikingly, in contrast to RUNX2 protein expression, which peaks in stage 1 and stage 2 progenitors, *RUNX2* transcript expression is

the highest in HSC and declines during NK cell differentiation. Also in several other studies, it has frequently been found that mRNA levels do not always mirror protein levels, which is probably due to differential translation efficiency, protein stability, and/or post-translational modifications in different cell subpopulations.

By introducing RUNX2(-I) knockdown as well as overexpression in human HSC-based NK cell differentiation cultures, we discovered that RUNX2 is important for generating stage 3 progenitors as well as stage 4 and stage 5 NK cells. RUNX2-I overexpression accelerates NK development significantly, since stage 4 NK cells are already detected at day 7 of culture, which takes at least 14 days to develop in control cultures. In addition, generation of stage 5 NK cells advances more rapidly compared to control conditions. Inversely, RUNX2 knockdown results in reduced stage 3–5 cell numbers. Despite earlier reports on the involvement of RUNX proteins in apoptosis and/or proliferation (*Mevel et al., 2019*; *Matthijssens et al., 2021*; *Coffman, 2003*), the altered NK cell development in RUNX2(-I) overexpression and knockdown cultures is not caused by distortions in these processes. Another possibility is that NK cell differentiation itself is affected. Research has shown that stage 3 progenitors are heterogeneous as they not only give rise to NK cells, but to ILC3s as well (*Björkström et al., 2016*). Commitment to the NK lineage is achieved when stage 3 progenitors express IL-2Rβ and thereby become sensitive to IL-15, an essential cytokine in NK cell development (*Geiger and Sun, 2016*; *Wang and Malarkannan, 2020*). Moreover, regulation of IL-2Rβ expression by Runx proteins has already been described in mouse models. Murine HSCs expressing a dominant-negative form of all Runx proteins generate less IL-2Rβ$^+$ NK cells compared to controls (*Ohno et al., 2008*) and in *Runx3*$^{-/-}$ mice, the responsiveness of NK cells to IL-15/IL-15Rα complexes is hindered (*Levanon et al., 2014*). Furthermore, pan-Runx ChIP-seq analysis has indicated that Runx proteins bind to the *Il2rb* promoter in murine NK cells (*Ohno et al., 2008*; *Levanon et al., 2014*). In line with these findings, we confirm using ChIP-seq that the *IL2RB* is a direct target of RUNX2 in human PB NK cells. Whereas overexpression prematurely gives rise to IL-2Rβ$^+$ stage 3 progenitors on day 7 of culture and increases IL-2Rβ levels in ALL-SIL cells, silencing does not generate a significant difference, which can be attributed to residual RUNX2 levels that might be sufficient to control IL-2Rβ expression. In addition on day 14 of culture, the frequency of IL-2Rβ$^+$ stage 3 progenitors is unaltered. However, this is not a counterargument for a regulating role of RUNX2 in IL-2Rβ expression as the IL-2Rβ$^+$ NK-committed progenitors from day 7 will probably have differentiated into stage 4 or stage 5 NK cells on day 14. This is in agreement with the increased absolute cell numbers of stage 4 and stage 5 NK cells in the RUNX2 overexpression cultures on day 14. Taken together, we deduce from these data that RUNX2 promotes NK cell development in part by inducing IL-2Rβ expression and thereby enabling NK lineage commitment.

Next, the molecular mechanism by which RUNX2 controls transcription was investigated in more detail. It is well known that RUNX2 by itself has a low affinity for DNA and therefore needs to form a transcriptional complex to increase efficiency (*Mevel et al., 2019*). HOMER analysis of RUNX2 ChIP-seq data of human PB NK cells reveals that genomic regions of RUNX2-specific peaks often contained *ETS1* and *TBX21* motifs. This is supported by our previous findings that, inversely, *RUNX* motifs are enriched in ETS1 ChIP-seq peaks of human NK cells (*Taveirne et al., 2020*). Interestingly, similar co-localisation of *Runx*, *ETS,* and *T-box* motifs has also been observed by others in murine NK cells (*Ebihara et al., 2015*; *Levanon et al., 2014*). These results strongly suggest that RUNX2 collaborates with ETS1 and T-box factors to govern developmental and/or functional programs in NK cells. The importance of RUNX2 as a transcriptional regulator is further highlighted by our RNA-seq analysis, performed on RUNX2-silenced or overexpressing human NK cells. Manipulating RUNX2 during NK development results in differential expression of a vast number of genes, the majority of which are direct targets of the transcriptional complex containing RUNX2.

Until recently, human NK cells were classified into two subsets, namely cytokine-producing CD56$^{bright}$CD16$^-$ and cytotoxic CD56$^{dim}$CD16$^+$ NK cells (*Abel et al., 2018*). While CD56$^{dim}$ NK cells are abundant in the circulation, CD56$^{bright}$ NK cells preferably reside in peripheral tissues (*Melsen et al., 2016*). Interestingly, studies with matched liver transplants and PB samples have demonstrated that there are CD56$^{bright}$ NK cells that do not recirculate and remain resident in the liver for years (*Cuff et al., 2016*). Similar trNK subsets have been described in the uterus, lungs, bone marrow, secondary lymphoid tissues, thymus, skin, kidneys, gut, and adipose tissue (*Valero-Pacheco and Beaulieu, 2020*). trNK cells are retained in these tissues by the expression of molecules such as CD69, CD49a, CXCR6,

CD103, and CCR5, that recruit and tether them to the stroma. In addition, they are prevented from migrating into the PB by the lack of circulatory-specific markers including CD49e, CX3CR1, CCR7, CD62L, S1PR1, and S1PR5 (*Vitale et al., 2020*; *Castriconi et al., 2018*; *Hashemi and Malarkannan, 2020*). Circulatory and resident NK cells are also known to have a different transcription factor profile, as EOMES^high T-BET^low characterises trNK cell subsets, and EOMES^low T-BET^high specifies circulatory NK cells (*Hashemi and Malarkannan, 2020*; *Collins et al., 2017*). Here we show that human NK cells from overexpression cultures increase expression of tissue-resident markers (CD69 and CD49a), while expression of circulatory factors (CD49e, CX3CR1, CD62L) is decreased. Moreover, the frequency of EOMES^high T-BET^low NK cells is increased in overexpression cultures. NK cells from knockdown cultures generally yield inverse trends. We also compared the differentially expressed genes from the RNA-seq results of RUNX2(-I) overexpression or knockdown NK cells with published gene sets, comprising up- and downregulated genes in liver- and bone marrow-resident versus recirculating NK cells (*Cuff et al., 2016*; *Melsen et al., 2018*). GSEA reveals that upregulated genes of NK cells from RUNX2-I overexpression cultures are highly enriched for the tissue-resident signature genes of both gene sets. In contrast, upregulated genes in RUNX2 knockdown NK cells are highly enriched for recirculating NK cell genes. In short, we proved that RUNX2 programs NK cells to acquire a tissue-resident phenotype.

Despite numerous shared features, some differences between resident NK cells of distinct host organs still exist, which are thought to be caused by the tissue-specific microenvironment. The profound impact of the microenvironment on the NK cell phenotype is evidenced by experiments showing that circulatory NK cells acquire tissue-resident markers upon ex vivo stimulation with tissue-specific conditioned medium (*Keskin et al., 2007*; *Harmon et al., 2016*). Each organ has a unique function and is home to a wide variety of self and non-self antigens. As a result, each organ has its own requirements regarding immunosurveillance and actively shapes immune cells to fit these needs (*Björkström et al., 2016*; *Jabrane-Ferrat, 2019*). It is therefore not surprising that not only their phenotype, but also the functionality of resident NK cells differs. For example, because the liver is constantly supplied with 'foreign' antigens originating from food components and microbiota in the gut, it is important that resident immune cells remain tolerant whilst still being responsive to potential threats. So, it is hypothesised that while retaining their cytotoxic potential, liver-resident NK cells are educated to be tolerogenic, which is reflected by reduced production of perforin, granzymes, IFN-γ, and TNF-α (*Cuff et al., 2016*; *Harmon et al., 2016*; *Stegmann et al., 2016*; *Zhao et al., 2020*; *Hudspeth et al., 2016*; *Hydes et al., 2018*; *Angelo et al., 2019*). A similar situation occurs in decidual NK cells of pregnant women, which come in close contact with factors originating from the growing fetus (*Jabrane-Ferrat, 2019*; *Kalkunte et al., 2008*). Although these NK cells produce cytokines and effector molecules to the same degree as their circulating counterparts, they are less cytotoxic (*Keskin et al., 2007*; *Koopman et al., 2003*). The tolerogenic phenotype of trNK cells is also evidenced by the increased expression of inhibitory receptor CD94/NKG2A (*Freud et al., 2017*; *Hashemi and Malarkannan, 2020*; *Jabrane-Ferrat, 2019*). However, KIR expression is not as consistent since liver- and decidua-resident NK cells have decreased and increased expression levels, respectively (*Freud et al., 2017*; *Peng and Tian, 2017*). Despite not affecting cytotoxicity, RUNX2-I overexpression generates tolerogenic NK cells with decreased production and/or secretion of perforin, granzyme B, IFN-γ, and TNF-α. Moreover, these findings are further validated by RUNX2 knockdown, which results in the opposite. Interestingly, although T-BET and EOMES expression are unaltered, silencing RUNX2 results in the upregulation of granzyme B and perforin. This is unexpected since EOMES and T-BET are known to promote expression of these cytotoxic effector molecules (*Cruz-Guilloty et al., 2009*). Our findings are therefore indicative of a T-BET- and/or EOMES-independent regulation of granzyme B and perforin expression by RUNX2. The RUNX2-specific ChIP-seq data indeed indicate that RUNX2 can directly regulate transcription of granzyme B and perforin. Additionally, RUNX2-I overexpression significantly increases inhibitory NK cell receptors CD94 and NKG2A, while decreasing the frequency of KIR^+ NK cells. The expression of activating NK receptors NKp44, NKp46, and NKG2C is also significantly affected by RUNX2 manipulation and has been detected by others in specific trNK cell subsets (*Cuff et al., 2016*; *Harmon et al., 2016*; *Koopman et al., 2003*; *Aw Yeang et al., 2017*). However, their exact function in tissue-resident NK cell subsets still remains unknown. In summary, due to the large variety of microenvironments in the body, it is difficult to study what drives specific trNK phenotypes and/or functions, especially using in vitro models. Nevertheless, we show that many characteristics of RUNX2-I-overexpressing NK cells, are also typical for liver-, decidual-, and/or bone marrow-resident

NK cells while knockdown cultures generate NK cells displaying circulatory features (*Hashemi and Malarkannan, 2020*; *Mikulak et al., 2019*).

All of the data discussed so far were generated using in vitro differentiation cultures. So, to validate the importance of RUNX2 in vivo, we examined the impact of RUNX2 knockdown on human NK cell development and tissue residency in humanised *NSG-huIL-15* mice. Silencing RUNX2 in human HSC leads to a severe NK cell deficiency in all examined organs of humanised *NSG-huIL-15* mice, which is evidence for an important role of RUNX2 in human NK cell development in vivo and confirms our in vitro data. In addition, we demonstrate that RUNX2 is required for the development and/or homing of human trNK cells in murine bone marrow and intestinal LPL. Interestingly, this is not true for every organ, since the frequency of liver and lung tr- and circNK cells is unaffected by RUNX2 knockdown. Moreover, the proportion of splenic circNK cells is decreased, which is in contradiction with NK cell frequencies in bone marrow and intestinal LPL. This suggests that either RUNX2-mediated regulation of NK cell homing or development is tissue-specific or that, at least for some organs, this mouse model is not suitable to study these processes in human. This might be due to human-mouse incompatibility of chemokines, chemokine receptors, and/or adhesion molecules. This could also explain the discrepancy in frequency of liver trNK cells between the humanised mouse model (±20% of human NK cells are tissue-resident in the mouse liver) and ex vivo human liver cells (±50% of human NK are tissue-resident; *Melsen et al., 2016*).

Circulating NK cells are important effectors against blood-borne tumours. However, elimination of solid tumours requires efficient homing and infiltration into the tumour microenvironment (*Ben-Shmuel et al., 2020*; *Nixon and Li, 2018*). Research has shown that this is partially enabled by the chemokine receptor CXCR4 (*Castriconi et al., 2018*; *Zavidij et al., 2020*). Additionally, it was demonstrated that NK cells in solid tumours share multiple characteristics with trNK cells (*Bruno et al., 2014*). For instance, similar to decidual NK cells, tumour-infiltrating and -associated NK cells produce angiogenic markers, which stimulate the de novo generation and remodelling of blood vessels and thus ensure blood supply to the rapidly growing tumour (*Bruno et al., 2014*). Tumour-infiltrating NK cells also resemble liver- and/or decidual NK cells in their reduced effector molecule and cytokine production in combination with diminished cytotoxicity (*Nixon and Li, 2018*; *Bruno et al., 2014*). So, increased CXCR4 levels in combination with the trNK cell phenotype observed in RUNX2-I overexpression cultures support a hypothetical role of RUNX2 in NK cell tumour infiltration.

In conclusion, we have shown that predominantly the RUNX2-I isoform is expressed during human NK cell development and that it sustains this process, in part by upregulating IL-2Rβ expression in NK cell-committed progenitors. In addition, we demonstrate, for the first time, that RUNX2 drives a tolerogenic tissue-resident phenotype in differentiating human NK cells. This not only improves our fundamental understanding of human NK cell biology, but it also creates opportunities for the potential use of in vitro NK cell differentiation models in generating NK cells that have enhanced tumour-infiltrating capabilities, to use in adoptive cell therapy for solid cancer patients.

## Materials and methods
### Cell lines

EL08-1D2 cells were kindly provided by E. Dzierzak (Erasmus University MC, Rotterdam, The Netherlands) and maintained in 50% Myelocult M5300 medium (Stem Cell Technologies, Grenoble, France), 35% α-MEM, 15% fetal calf serum (FCS; Biowest, Nuaillé, France), penicillin (100 U/mL), streptomycin (100 µg/mL), glutamine (2 mM; all from Life Technologies), and 10 µM β-mercaptoethanol on gelatine-coated (0.1%) plates at 33°C, 5% $CO_2$. For inactivation, EL08-1D2 cells were exposed to mitomycin C (10 µg/mL; Sigma-Aldrich, St. Gallen, Switzerland) for 3 hr. After extensive rinsing, the cells were harvested using trypsin-EDTA (Lonza, Bazel, Switzerland). EL08-1D2 cells were plated at a density of 50,000 cells/24-well, coated with 0.1% gelatin at least 24 hr before seeding the HSC or cultured NK cells. **K562** cells (ATCC, Manassas, VA) were cultured at 37°C, 5% $CO_2$ in complete Iscove's Modified Dulbecco's Medium (IMDM, Life Technologies) supplemented with 10% FCS, penicillin (100 U/mL), streptomycin (100 µg/mL), and glutamine (2 mM; Life Technologies). **293T** cells (ATCC) were maintained at 37°C, 7% $CO_2$ in complete DMEM Glutamax supplemented with 10% FCS, penicillin (100 U/mL), streptomycin (100 µg/mL), and glutamine (2 mM). **Phoenix-A** cells (ATCC) were cultured at 37°C 7% $CO_2$ in complete IMDM supplemented with 10% FCS, penicillin (100 U/mL), streptomycin (100 µg/

mL), and glutamine (2 mM). **ALL-SIL** cells (DSMZ, Braunschweig, Germany) were cultured at 37°C 5% $CO_2$ in complete RPMI medium supplemented with 20% FCS, penicillin (100 U/mL), streptomycin (100 µg/mL), and glutamine (2 mM; Life Technologies). **YTS** cells (ATCC) were maintained at 37°C 7% $CO_2$ in complete IMDM medium supplemented with 15% FCS, penicillin (100 U/mL), streptomycin (100 µg/mL), glutamine (2 mM), HEPES (0.01 M; all from Life Technologies), and β-mercapthoethanol (50 µM; Sigma-Aldrich). The identities of the cell lines bought from ATCC or DSMZ were authenticated by STR profiling. The identity of the EL08-1D2 cells was verified by the donating research group. In addition, all cell lines tested negative for mycoplasma contamination.

## Tissue collection and cell isolation

All tissues were collected with approval by the Ethics Committee of the Faculty of Medicine and Health Sciences of Ghent University (ref#2017/0227, Ghent, Belgium). Informed consents were received according to the Declaration of Helsinki. **Human bone marrow** was obtained by bone marrow puncture. Thereafter, CD34$^+$CD19$^-$CD56$^-$ cells were sorted and used for analysis. **Human tonsils** were retrieved fresh from surgery and processed to single-cell suspensions. CD3$^-$CD19$^-$ mononuclear cells were enriched by Lymphoprep (Stem Cell Technologies, Grenoble, France) density gradient centrifugation followed by magnetic-activated cell sorting using biotin MicroBead kit (Miltenyi Biotec, Leiden, The Netherlands), according to manufacturer's guidelines. **Human umbilical cord blood (UCB) and PB** were obtained from the Blood Bank of Ghent University. Mononuclear cells were isolated using Lymphoprep density gradient centrifugation. Enrichment of UCB-derived CD34$^+$ precursors and PB-derived NK cells was accomplished by magnetic-activated cell sorting using the CD34 MicroBead kit and the NK cell isolation kit (Miltenyi Biotec), respectively.

## Viral constructs

The *RUNX2-specific shRNA* (5'-CTACCTATCACAGAGCAATT-3') was subcloned in the lentiviral vector (pLKO.1) containing *eGFP* as reporter gene. The *scrambled shRNA* pLKO.1 was used as a negative control. Lentivirus was generated by transfection in the 293T cell line using JetPEI (Polyplus transfection, Illkirch, France), as specified by the manufacturer. Viral supernatant was harvested 48 and 72 hr after transfection and stored at –80°C. RUNX2 has two principal isoforms, namely type-I and type-II RUNX2. cDNA of *Type-I RUNX2* was subcloned in the *LZRS-internal ribosome entry site* (*IRES*)-*eGFP* retroviral vector, containing the puromycin resistance gene. The empty *LZRS-IRES-eGFP* vector was used as negative control. Retrovirus was produced by transfection in the Phoenix-A-based amphotropic packaging cell line using calcium-phosphate precipitation (Thermo Fisher, Waltham, MA). Viral supernatant was collected 2, 6 and 14 days after transfection and stored at –80°C.

## UCB HSC-based NK cell differentiation cultures

After 48 hr of culture in preculture medium (complete IMDM medium supplemented with 10% FCS, stem cell factor [SCF; 100 ng/mL, Peprotech], FMS-like tyrosine kinase-3 ligand [FLT3-L; 100 ng/mL, R&D systems], and thrombopoietin [TPO; 20 ng/mL, Peprotech]), CB-CD34$^+$ precursors were transduced with lenti- (*RUNX2 shRNA* or control) or retrovirus (*Type-I RUNX2* overexpression or control) by spinning at 950 g for 90 min at 32°C. To enhance transduction efficiency, RetroNectin (12 µg/mL, Takara Bio, Saint-Germain-en-Laye, France) was coated and the previously mentioned cytokines were added in combination with polybrene (8 µg/mL, Sigma-Aldrich). The latter was only used in case of lentiviral transduction, and it was removed after 24 hr by replacing medium and cytokines. CD34$^+$CD45RA$^-$Lin (CD3, CD14, CD56, CD19)$^-$ eGFP$^+$ HSC were sorted 48 hr after transduction using the FACSAria III cell sorter (BD Biosciences, San Jose, CA) and cultured on a monolayer of mitomycin C-inactivated (Sigma-Aldrich) EL08-1D2 cells in NK coculture (NKCC) medium, which consists of DMEM (Life Technologies) and Ham's F12 medium (2:1 ratio), supplemented with penicillin (100 U/mL), streptomycin (100 µg/mL), glutamine (2 mM), sodium pyruvate (10 mM; Life Technologies), 20% heat-inactivated human AB serum (Biowest, Nuaillé, France), β-mercapthoethanol (24 µM), ascorbic acid (20 µg/mL), and sodium selenite (50 ng/mL; all from Sigma-Aldrich). To induce NK cell differentiation, IL-3 (5 ng/mL), IL-7 (20 ng/mL), IL-15 (10 ng/mL; all from R&D systems), SCF (20 ng/mL), and FLT3-L (10 ng/mL) were also included. On day 7, the volume of the well was doubled with medium containing the previously mentioned cytokines with the exception of IL-3. On day 14, cells were harvested and transferred to new inactivated EL08-1D2 feeder cells in fresh medium enriched with cytokines.

## Flow cytometry

Samples for analysis by flow cytometry and for cell sorting were run on the LSRII or FACSARIAII (BD Biosciences, San Jose) flow cytometer. FlowJo_V10 software (Ashland, OR) was used for analysis. The antibodies and kits used according to the manufacturer's guidelines are listed in *Table 1*.

## qPCR analysis

Cells were lysed in RLT PLUS buffer and stored at –80°C until RNA isolation with the RNeasy Micro Kit (Qiagen, Hilden, Germany). cDNA was generated using the iScript Advanced cDNA generation kit (Bio-Rad, Hercules, CA) according to the manufacturer's guidelines. Quantitative PCR was performed in duplicate with the LightCycler 480 SYBR Green I Master mix (Roche, Bazel, Switzerland) on the Light-Cycler 480 real-time PCR system (Roche). Relative gene expression was determined using *GAPDH* and either *TBP* or *actin-β* as housekeeping genes. Primer sequences are listed in *Table 2*. *RUNX2 isoform*-specific primers were validated with gBlocks of *RUNX2-I* and *RUNX2-II* principal isoforms, depicted in *Table 3*.

## CellTrace experiments

On day 17 of NK differentiation culture, cells were labelled using the CellTrace Violet Cell Proliferation Kit, for flow cytometry (Life Technologies, Waltham, MA), according to the manufacturer's guidelines. Thereafter, eGFP$^+$ CellTrace$^+$ NK cells (CD45$^+$CD94$^+$CD56$^+$) were sorted and recultured in NKCC medium with the previously mentioned cytokines. After 4 days, cells were stained for NK-specific markers, and proliferation was examined by flow cytometry.

## ChIP-sequencing

Viable human NK cells (PI$^-$CD45$^+$CD56$^+$CD3$^-$CD19$^-$) were isolated from PB. The cells were then lysed by snap freezing. NK cells were fixed with 1% formaldehyde for 15 min and quenched with 0.125 M glycine. Chromatin was isolated by addition of lysis buffer, followed by disruption with a Dounce homogeniser. Lysates were sonicated on a microtip sonicator and the DNA sheared to an average length of 300–500 bp. Control genomic DNA (further referred to as input DNA) was prepared by treating aliquots of chromatin with RNase, proteinase K, and heat for de-crosslinking, followed by ethanol precipitation. Pellets were resuspended and the resulting DNA was quantified on a NanoDrop spectrophotometer. Extrapolation to the original chromatin volume allowed quantitation of the total chromatin yield. An aliquot of chromatin (30 µg) was precleared with protein A agarose beads. Genomic DNA regions of interest were isolated using 10 µl of RUNX2 antibody (Cell Signalling Technologies, clone D1H7). Complexes were washed, eluted from the beads with SDS buffer, and subjected to RNase and proteinase K treatment. Crosslinks were reversed by incubation overnight at 65°C, and ChIP DNA was purified by phenol-chloroform extraction and ethanol precipitation. Quantitative PCR reactions were carried out in triplicate on specific genomic regions using SYBR Green Supermix (Bio-Rad, Hercules, CA). The resulting signals were normalised for primer efficiency by carrying out qPCR for each primer pair using input DNA. Illumina sequencing libraries were prepared from the ChIP and input DNAs by the standard consecutive enzymatic steps of end-polishing, dA-addition, and adaptor ligation. After a final PCR amplification step, the resulting DNA libraries were quantified and sequenced on Illumina's NextSeq 500 (75 nt reads, single end). QC on fastq files was performed with FastQCv0.11.7 (*Andrews, 2010*). Fastq files were aligned to hg38 using Burrows-Wheeler Aligner (BWA). Peak calling was done with MACS2 v2.1.0. The RUNX2 ChIP-seq peaks were analysed in IGV along with publicly available histone (H3K27ac and H3K4Me3) ChIP-seq and ATAC-seq data of human peripheral NK cells (GSE107147 and GSE77299; *Koues et al., 2016*). For H3K4Me3 and H3K27Ac, peaks were called from bedGraph files with the MACS2 peak calling software. First, MACS2 bdgcmp was used to compare ChIP and input files. Thereafter, peaks were called using MACS2 bdgpeakcall. Options for MACS2 bdgpeakcall were set to -c 2, -g 100 and -l 100 for H3K4Me3 or 150 for H3K27Ac. RUNX2 motifs in promoters and enhancers were identified using MACS2 and HOMER software (*Heinz et al., 2010*; *Zhang et al., 2008*).

## Library preparation, RNA-sequencing, and analysis

Human NK cells (eGFP$^+$CD45$^+$CD94$^+$CD56$^+$) of four to five replicates were sorted from RUNX2-I overexpression (d14) or RUNX2 knockdown (d21) cultures, or their respective controls. RNA was

**Table 1.** Antibodies and kits used in flow cytometric analysis.

**ANTIBODIES**

| Marker | Alternative | Fluorochrome | Clone | Supplier |
|---|---|---|---|---|
| Fc blocking agent (human) | / | / | / | Miltenyi Biotec, Leiden, The Netherlands |
| Anti-mouse FcγII/III | / | Unconjugated | 2.4G2 | Kindly provided by Dr. J. Unkeless, Mount Sinai School of Medicine, NY |
| CD34 | CD34 | Phycoerythrin (PE), Pacific Blue (PB) | AC136 | Miltenyi Biotec, Leiden, The Netherlands |
| CD45RA | CD45RA | Allophycocyanin (APC) | HI100 | Biolegend, San Diego, CA |
| CD45 | CD45 | Allophycocyanin/Fire 750 (APC-Fire) | 2D1 | Biolegend, San Diego, CA |
| CD117 | KIT | Phycoerythrin-Cyanin7 (PECy7) | 104D2 | Thermo Fischer Scientific, Waltham, MA |
| CD94 | KLRD1 | Peridinin Chlorophyll Protein-Cyanin5.5 (PerCP-Cy5,5) | DX22 | Biolegend, San Diego, CA |
| CD56 | NCAM1 | VioBlue (VB), Allophycocyanin (APC) | 5.1H11 | Biolegend, San Diego, CA |
| CD16 | FcγRIIIA | Phycoerythrin (PE), Allophycocyanin (APC) | B73.1 | Biolegend, San Diego, CA |
| CD3 | CD3 | Allophycocyanin (APC) | SK7 | BD Biosciences, San Jose, CA |
| CD14 | CD14 | Allophycocyanin (APC) | REA599 | Miltenyi Biotec, Leiden, The Netherlands |
| HLA-DR | HLA-DR | eFluor780 | LN3 | Thermo Fischer Scientific, Waltham, MA |
| NKp44 | CD336;NCR2 | eFluor450, Allophycocyanin (APC) | 44.189 | Thermo Fischer Scientific, Waltham, MA |
| CD19 | CD19 | Allophycocyanin (APC) | SJ25C1 | Thermo Fischer Scientific, Waltham, MA |
| CD122 | IL2RB | Phycoerythrin (PE) | MIKβ3 | BD Biosciences, San Jose, CA |
| EOMES | EOMES | Phycoerythrin (PE), Allophycocyanin (APC) | WD1928 | Thermo Fischer Scientific, Waltham, MA |
| T-BET | T-BET | Phycoerythrin (PE) | 4B10 | Thermo Fischer Scientific, Waltham, MA |
| HELIOS | IKZF2 | Allophycocyanin (APC) | 22F6 | Biolegend, San Diego, CA |
| PLZF | ZBTB16 | Phycoerythrin (PE) | R17-809 | BD Biosciences, San Jose, CA |
| CD107a | LAMP1 | Phycoerythrin (PE) | H4A3 | BD Biosciences, San Jose, CA |
| Granzyme B | Granzyme B | Phycoerythrin (PE) | GB11 | Thermo Fischer Scientific, Waltham, MA |
| Perforin | Perforin | Phycoerythrin (PE) | dG9 | Thermo Fischer Scientific, Waltham, MA |
| IFN-γ | IFN-γ | eFluor 660 | 4 S.B3 | Thermo Fischer Scientific, Waltham, MA |
| TNF-α | TNF-α | Phycoerythrin (PE) | Mab11 | Thermo Fischer Scientific, Waltham, MA |
| RUNX1 | AML1, CBFA2 | Phycoerythrin (PE) | RXDMC | Thermo Fischer Scientific, Waltham, MA |
| RUNX2 | AML3, CBFA1 | Phycoerythrin (PE) | D1L7F | Cell Signalling Technologies, Leiden, The Netherlands |
| RUNX3 | AML2, CBFA3 | Phycoerythrin (PE) | R3-5G4 | BD Biosciences, San Jose, CA |
| NKp46 | CD335;NCR1 | Phycoerythrin-Cyanin7 (PECy7) | 9E2 | Biolegend, San Diego, CA |
| NKG2C | CD159c | Phycoerythrin (PE) | FAB138P | R&D systems, Minneapolis MN |
| NKG2A | CD159a | Allophycocyanin (APC) | REA110 | Miltenyi Biotec, Leiden, The Netherlands |
| CD158a,h | KIR2DL1/KIR2DS1 | Phycoerythrin (PE) | 3B6.B | Beckman Coulter, Brea, CA |
| CD158b1/b2 | KIR2DL2/KIR2DL3 | Phycoerythrin (PE) | GL183 | Beckman Coulter, Brea, CA |
| CD158i | KIR4DS1 | Phycoerythrin (PE) | FES172 | Beckman Coulter, Brea, CA |
| CD158e1/e2 | KIR3DL1/KIR3DS1 | Phycoerythrin (PE) | Z27.3.7 | Beckman Coulter, Brea, CA |
| CD69 | CD69 | Phycoerythrin (PE) | FN50 | Biolegend, San Diego, CA |
| CD49a | ITGA1 | Alexa Fluor 647 | TS2/7 | Biolegend, San Diego, CA |
| CD49e | ITGA5 | Allophycocyanin (APC) | IIA1 | BD Biosciences, San Jose, CA |
| CX3CR1 | V28, GPR13 | Phycoerythrin (PE) | 2A9-1 | Biolegend, San Diego, CA |

*Table 1 continued on next page*

*Table 1 continued*

**ANTIBODIES**

| Marker | Alternative | Fluorochrome | Clone | Supplier |
|---|---|---|---|---|
| CCR7 | CD197 | Allophycocyanin/Fire 750 (APC-Fire) | G043H7 | Biolegend, San Diego, CA |
| CD62L | L-selectin | Allophycocyanin (APC) | DREG-56 | Biolegend, San Diego, CA |
| S1PR1 | CD363 | eFluor660 | SW4GYPP | Thermo Fischer Scientific, Waltham, MA |
| CXCR4 | CD184 | Phycoerythrin (PE) | 12G5 | BD Biosciences, San Jose, CA |
| Ki67 | Ki67 | Phycoerythrin (PE) | SolA15 | Thermo Fischer Scientific, Waltham, MA |
| Streptavidin | Streptavidin | Allophycocyanin (APC) | / | BD Biosciences, San Jose, CA |
| Fixable viability dye | Fixable viability dye | eFluor506 | / | Thermo Fischer Scientific, Waltham, MA |

**KITS**

| | | | | |
|---|---|---|---|---|
| AnnexinV apoptosis detection kit | | Allophycocyanin (APC) | / | Thermo Fischer Scientific, Waltham, MA |
| FoxP3/Transcription Factor Staining Buffer set | | / | / | Thermo Fischer Scientific, Waltham, MA |

extracted from sorted NK cells using the RNeasy Micro Kit (Qiagen, Hilden, Germany). Concentration and quality of the RNA were checked using the 'Quant-it ribogreen RNA assay' (Life Technologies, Grand Island, NY) and the RNA 6000 nano chip (Aligent Technologies, Santa Clara, CA), respectively. The QuantSeq 3' mRNA-Seq Library Prep Kit (Lexogen, Vienna, Austria) was used to perform an Illumina sequencing library preparation on 22 and 42 ng of RNA of NK cells from RUNX2 knockdown and RUNX2-I overexpression cultures, respectively. These libraries were quantified by qPCR, as reported by Illumina's protocol 'Sequencing Library qPCR Quantification protocol guide', version February 2011. To control the library's size distribution and quality, a high-sensitivity DNA chip (Agilent Technologies, Santa Clara, CA) was used, after which sequencing was performed on a high throughput Illumina NextSeq 500 flow cell generating 75 bp single reads. An average of $8.0 \times 10^6 \pm 0.2 \times 10^6$ reads and $5.0 \times 10^6 \pm 0.9 \times 10^6$ reads were generated per sample of RUNX2 knockdown and RUNX2-I overexpression cultures, respectively. Quality control was performed on fastq files with FasQCv0.11.7 (*Andrews, 2010*). Fastq files were aligned to hg38 using STAR2.42 and genes (Gencode v25) were quantified on the fly. Differential expression analysis was done in R with DESeq2 (*Love et al., 2014*) with Wald test. Genes with an FDR <0.1 were considered significantly differential.

**Table 2.** qPCR primers.

| Gene | Sense | Sequence (5' → 3') |
|---|---|---|
| | Sense | ATGCGTATTCCCGTAGATCC |
| RUNX2-I | Antisense | GGGCTCACGTCGCTCATTT |
| | Sense | AGGAGGGACTATGGCATCAAAC |
| RUNX2-II | Antisense | GGGCTCACGTCGCTCATTT |
| RUNX2 (flanking shRNA binding site) | Sense | CACCACTCACTACCACACCT |
| | Antisense | AGCATTCTGGAAGGAGACCG |
| | Sense | TCCTCTGACTTCAACAGCGACA |
| GAPDH | Antisense | GTGGTCGTTGAGGGCAATG |
| | Sense | CACGAACCACGGCACTGATT |
| TBP | Antisense | TTTTCTTGCTGCCAGTCTGGAC |
| | Sense | ATGACCCAGATCATGTTTGAGA |
| ACTB | Antisense | AGAGGCGTACAGGGATAGCA |

**Table 3.** gBlock sequences of RUNX2 isoforms.

| gBlock | Sequence |
|---|---|
| RUNX2-I | GTCTCGCCTTCACCCCCCCAATTTCCTCCTTGCCCCTCATTTCCACCCTCCTCCCCCGGCCACTTGGCTGTTGTGATGC GTATTCCCGTAGATCCGAGCACCAGCCGGGCGCTTCAGCCCCCCCCCCCCCGCAGCCCGGCAAAATGAGCGACGTGAGCCCGGTGGTG GCTGCG |
| RUNX2-II | CAAACTTTCTCCAGGAGGACAGCAAGAAGTCTCTGGTTTTTAAAATGGTTAATCTCCGCAGGTCACTACCAGCCACCGAGACCAACAGAGTCAGTGA GTGCTCTCTAACCACAGTCTATGCAGTAATAGTAGGTCCTTCAAATATTTGCTCATTCTCTTTTGTTTCTTTGCTTTTCACATGTTACCAGCTA CATAATTTCTTGACAGAAAAAATAAATAAAAGTCTATGTACTCCAGGCATACTGTAAAACTAAAACAAGGTTTGGGTATGGTTTGTATTTCAGTTTA AGGCTGCAAGCAGTATTTACAACAGAGGGTACAAGTTCTATCTGAAAAAAAAGGAGGGACTATGGCATCAAACAGCCTCTTCAGCACAGTGACAC CATGTCAGCAAAACTTCTTTTGGGGATCCGAGCACCAGCCGGCGCTTCAGCCCCCTCCAGCGCCTGCAGCCTGCAGCCCGGCAAAATGAGCGACGTG AGCCCGGTGGTGGCTG |

## GSEA analysis

GSEA was performed using GSEA software tool v4.1.0 of the Broad Institute (*Mootha et al., 2003*; *Subramanian et al., 2005*). The 'GSEAPreranked' module was run using standard parameters and 1000 permutations. Datasets contained differentially expressed genes of NK cells from either RUNX2 knock-down (d21) or RUNX2-I overexpression (d14) cultures versus their respective controls. The liver gene set (GSE87392) comprised the top 500 significantly up- or downregulated genes of resident (CD69+CX-CR6+EOMES$^{high}$) versus recirculating (CD69−CXCR6−EOMES$^{low}$) NK cells (*Cuff et al., 2016*). The bone marrow gene set (GSE116178) consisted of all significantly up- or downregulated genes of resident CD56$^{bright}$ (CD69+CXCR6+) versus recirculating CD56$^{bright}$ NK cells (CD69−CXCR6−; *Melsen et al., 2018*).

## Functional assays

### $^{51}$Chromium release assay

K562 target cells were labelled with $Na_2CrO_4$ (*Bruno et al., 2014*; Perkin Elmer, Waltham, MA). eGFP+ NK cells (CD45+CD94+CD56+) were sorted and incubated in the presence of *Bruno et al., 2014* Cr-labeled K562 cells at variable effector-target ratio's (E:T) in triplicate for 4 hr. Thereafter, the supernatant was mixed with scintillation fluid and the signal was measured using a 1450 LSC&Luminescence Counter (Wallac Microbeta Trilux, Perkin Elmer). The mean percentage of specific lysis was determined.

### Degranulation assay

Cultured cells were harvested and co-cultured in bulk with K562 target cells in NKCC medium at a 1:1 ratio for 2 hr. Thereafter, cells were stained with CD56, CD94, and CD107a antibodies. The presentation of CD107a+ NK cells was measured by flow cytometry.

### Cytokine stimulation

Cultured cells were harvested and stimulated in bulk in NKCC medium supplemented with either PMA (5 ng/mL) and ionomycin (1 μg/mL) for 6 hr, or with IL-12/IL-18 (both 10 ng/mL) or with IL-12/IL-18/IL-15 (latter 4 ng/mL) for 24 hr. Brefeldin A (BD GolgiPlug, BD Biosciences) was added 4 hr prior to harvesting. NK cell-specific surface markers were stained and cells were fixed and permeabilised using the Cytofix/Cytoperm Kit (BD Biosciences) before adding anti-IFN-γ and anti-TNF-α. To investigate cytokine secretion, sorted eGFP+ NK cells (CD45+CD94+CD56+) were exposed to the previously described stimuli for 24 hr, after which supernatant was collected and analysed with IFN-γ ELISA assay (PeliKine-Tool Set, Sanquin, Amsterdam, The Netherlands).

### Humanised mouse model

All animal experiments were performed after approval and in accordance with the guidelines of the Ethical Committee for Experimental Animals at the Faculty of Medicine and Health Sciences of Ghent University (ref# ECD20/20, Ghent, Belgium). CD34+ HPC were isolated by magnetic-activated cell sorting using the CD34 MicroBead kit (Miltenyi Biotec) and cultured for 16 hr in preculture medium, after which cells were transduced with lentivirus (*RUNX2 shRNA* or control), as described in a previous section ('*UCB HSC-based NK cell differentiation cultures*'). At 4 hr after transduction, bulk HPC were intravenously injected in lethally irradiated (200cGy) *NOD SCID gamma* mice, which were transgenic for human IL-15 (*NSG-huIL–15*). These mice were kindly given by B. Vandekerckhove. At 6–7 weeks post-injection, the mice were sacrificed by cervical dislocation and perfused with PBS. Single-cell suspensions of liver, spleen, bone marrow, and intestinal LPL were generated as previously described (*Van Acker et al., 2017*; *Filtjens et al., 2016*; *Filtjens et al., 2013*). Lungs were cut in pieces and digested in RPMI1640 (Life Technologies) supplemented with 2% FCS, collagenase D (2 mg/mL, Roche), and Dnase I (0.2 mg/mL, Roche). Thereafter, single-cell suspension was prepared using Percoll (VWR, Radnor, PA), and red blood cells were lysed with ACK lysis buffer (Life Technologies). The presence of human NK cells (CD45+CD56+CD94+) and the frequency of human tissue-resident (CD69+CD49e-) and circulating (CD69-CD49e+) NK cells were analysed using flow cytometry.

## Statistical analysis

Statistical significance of the in vitro experiments was determined with the Student's ratio paired t-test, while the significance of the in vivo data was determined with the Student's unpaired t-test. The

statistical tests were performed with GraphPad Prism version 9.0.2 for Windows (GraphPad Software, San Diego, CA). A p-value <0.05 was considered statistically significant.

## Data sharing statement

ChIP-seq and RNA-seq data are accessible on GEO (accession number 172439). For original data, please contact georges.leclercq@ugent.be.

## Acknowledgements

This work was supported by grants from the Research Foundation of Flanders (FWO) (grants G.0444.17N, 1S45317N, 12N4515N, 1S29317N) (G L, S W, S T and L K, respectively) and Kinder-kankerfonds (a non-profit childhood cancer foundation under Belgian law to P V V and W V L). The computational resources (Stevin Supercomputer Infrastructure) and services used in this work were provided by the VSC (Flemish Supercomputer Center), funded by Ghent University, FWO and the Flemish Government – department EWI. Practical expertise and assistance regarding ChIP-seq and RNA-seq used in this study was provided by Active Motif (Carlsbad, CA) and NXTGNT (Ghent, Belgium), respectively.

## Additional information

### Funding

| Funder | Grant reference number | Author |
|---|---|---|
| Fonds Wetenschappelijk Onderzoek | G.0444.17N | Georges Leclercq |
| Fonds Wetenschappelijk Onderzoek | 1S45317N | Sigrid Wahlen |
| Fonds Wetenschappelijk Onderzoek | 12N4515N | Sylvie Taveirne |
| Fonds Wetenschappelijk Onderzoek | 1S29317N | Laura Kiekens |

The funders had no role in study design, data collection and interpretation, or the decision to submit the work for publication.

### Author contributions

Sigrid Wahlen, Conceptualization, Data curation, Formal analysis, Funding acquisition, Investigation, Methodology, Project administration, Validation, Visualization, Writing - original draft, Writing – review and editing; Filip Matthijssens, Pieter Van Vlierberghe, Resources, Writing – review and editing; Wouter Van Loocke, Data curation, Formal analysis, Software, Writing – review and editing; Sylvie Taveirne, Supervision, Writing – review and editing; Laura Kiekens, Eva Persyn, Stijn De Munter, Investigation, Writing – review and editing; Els Van Ammel, Zenzi De Vos, Investigation, Methodology, Writing – review and editing; Patrick Matthys, Tom Taghon, Writing – review and editing; Filip Van Nieuwerburgh, Formal analysis, Investigation, Writing – review and editing; Bart Vandekerckhove, Formal analysis, Investigation, Methodology, Resources, Writing – review and editing; Georges Leclercq, Conceptualization, Funding acquisition, Methodology, Supervision, Writing – review and editing

### Author ORCIDs

Sigrid Wahlen http://orcid.org/0000-0003-2606-1023
Stijn De Munter http://orcid.org/0000-0003-3821-0620
Patrick Matthys http://orcid.org/0000-0002-9685-6836
Tom Taghon http://orcid.org/0000-0002-5781-0288
Pieter Van Vlierberghe http://orcid.org/0000-0001-9063-7205
Georges Leclercq http://orcid.org/0000-0002-1691-5294

## Ethics

All tissues were collected with approval by the Ethics Committee of the Faculty of Medicine and Health Sciences of Ghent University (ref#2017/0227, Ghent, Belgium). Informed consents were received according to the Declaration of Helsinki.

All animal experiments were performed after approval and in accordance with the guidelines of the Ethical Committee for Experimental Animals at the Faculty of Medicine and Health Sciences of Ghent University (ref# ECD20/20, Ghent, Belgium).

## Decision letter and Author response

Decision letter https://doi.org/10.7554/eLife.80320.sa1
Author response https://doi.org/10.7554/eLife.80320.sa2

---

# Additional files

## Supplementary files

- MDAR checklist
- Supplementary file 1. RUNX2-specific ChIP-seq data of human peripheral blood NK cells.
- Supplementary file 2. RNA-seq data of sorted human NK cells from RUNX2 knockdown cultures.
- Supplementary file 3. RNA-seq data of sorted human NK cells from RUNX2-I overexpression cultures.
- Supplementary file 4. Overlap of RNA-seq and ChIP-seq data sets.
- Supplementary file 5. GSEA analysis leading edge genes.

## Data availability

RNA-sequencing data and ChIP-sequencing data have been deposited in GEO under accession code GSE172439.

The following dataset was generated:

| Author(s) | Year | Dataset title | Dataset URL | Database and Identifier |
|---|---|---|---|---|
| Leclercq G | 2022 | The transcription factor RUNX2 drives the generation of human NK cells and promotes tissue residency | http://www.ncbi.nlm.nih.gov/geo/query/acc.cgi?acc=GSE172439 | NCBI Gene Expression Omnibus, GSE172439 |

The following previously published datasets were used:

| Author(s) | Year | Dataset title | Dataset URL | Database and Identifier |
|---|---|---|---|---|
| Cuff AO | 2016 | Comparison of Eomes-negative and Eomes-positive human liver NK cells by RNASeq | https://www.ncbi.nlm.nih.gov/geo/query/acc.cgi?acc=GSE87392 | NCBI Gene Expression Omnibus, GSE87392 |
| Melsen J | 2018 | Human bone marrow resident natural killer cells have a unique transcriptional profile and resemble resident memory CD8+ T cells | https://www.ncbi.nlm.nih.gov/geo/query/acc.cgi?acc=GSE116178 | NCBI Gene Expression Omnibus, GSE116178 |

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
