## [Editor Report]

This study provides new insights into the role of RUNX2 in the development and function of human NK cells. The impact of RUNX2 has been examined in in vitro models and in humanized mouse models to identify the pathways that might be impacted in vivo. The authors uncover a number of intriguing observations that impact the regulation of tissue-resident human NK cells identifying RUNX2 as another player in the regulation of NK cells.

---

## [Decision Letter]

[Editors' note: this paper was reviewed by Review Commons.]

---

## [Author Response]

Reviewer #1Summary:The manuscript titled "the transcription factor RUNX2 promotes the development of human tissue-resident NK cells" addressed a new role of RUNX2 on human NK cell development and phenotypes. It also provides a new insight into how RUNX2 affects human NK cells switch between the circulatory and tissue-resident NK cellsMajor comments:1. The authors described that IL-2Rb and other NK cell receptors are not affected in the RUNX2 shRNA system. However, the evidence is not enough to conclude that RUNX2 controls human NK cell development by direct induction of IL-2Rb expression, for the knockdown system did affect the Granzyme B and perforin expression without affecting EOMES and T-bet. More direct evidence should be provided to address the knockdown or knockout system that has a direct induction of IL-2Rb on NK cell development.

You are indeed correct that our conclusion regarding the direct RUNX2-mediated regulation of IL2Rβ was overstated and that additional experiments needed to be performed in order to validate our strong claims. We tried to demonstrate the direct induction of *IL-2Rβ* by RUNX2 using the IL2Rβ reporter construct of Genecopoeia (HPRM30810-LvPM03 with IL-2Rβ promoter sequence, 1585 bp in length, 269bp downstream of TSS). We introduced both this IL-2Rβ reporter vector and the RUNX2-I or empty control LZRS vector in the RUNX2^low^ IL-2Rβ^low^ ALL-SIL cell line to examine whether RUNX2 is able to activate the promoter of IL-2Rβ. Unfortunately, we were unable to generate conclusive results. Since we are unable to definitively demonstrate the direct induction of IL-2Rβ by RUNX2, we have nuanced our statements in the manuscript. Our data still show that RUNX2 affects IL-2Rβ expression in human NK cell development in vitro. We adjusted the text in the manuscript.

2. In Figure 1e, n is 4-12 for the experiments. Could the authors provide the reason for that? This may increase the bias to use sample number 4 vs. 12. For example, shRNA system d14 of NK cell result may not reflect the truth by 4 vs. 12

The data shown in Figure 1e are the result of multiple experiments using NK cell differentiation cultures, which were performed with transduced umbilical cord blood-derived hematopoietic stem cells (HSC). Whereas the number of samples was indeed different at the different analysis timepoints, the data of the paired control and RUNX2 shRNA/RUNX2-I graphs contain equal sample numbers. So, e.g. both control and RUNX2 shRNA HSC: n=4, while both control and RUNX2 shRNA NK d21: n=12. To illustrate this, I show all individual donor data in Author response image 1 as information for the reviewer.

**Author response image 1. sa2fig1:** 

Minor comments:1. Figure 1a shows the protein expression level of RUNX 2. Figure 1d shows the qPCR level of RUNX2-1. Could more explanation be provided that ST1 and ST2 have lower mRNA levels but higher protein levels than the d0?

You are indeed correct that the expression levels of RUNX2 in human NK cell developmental stages differ between protein and mRNA levels. However, mRNA levels do not always mirror protein levels, as has also been frequently found in several other studies. Possible causes for this phenomenon include different translation efficiency, protein stability and/or posttranslational modification in different subpopulations of cells. I have included this explanation in the discussion (page 10 line 239-243).

2. Please check the CCR7 flow data in Figure 4A circulatory. The histogram did not reflect the bar plot result. It looks like the expression of CCR4 between Ctrl and RUNX2-1 is different.

Thank you for this remark. I re-examined the data concerning CCR7 and included a more representative biological replicate in the revised figure 4a.

3. Please provide more discussion or explanation that the EOMES and T-bet level are not changed in the knockdown system, but the Granzyme B and perforin expression is changed there.

You are indeed correct that despite the unaltered expression T-BET and EOMES, two transcriptional regulators of granzyme B and perforin, RUNX2 knockdown results in the upregulation of these cytotoxic effector molecules. However, as shown by the RUNX2-specific ChIP-seq, granzyme B and perforin are direct targets of RUNX2, which indicates that the expression of these cytotoxic effector molecules can be directly regulated by RUNX2, independent of T-BET and EOMES. I have included this insight in the discussion (page 13 line 338-343).

4. Figure 5 only showed CD107a+ percentage of NK; how about the CD107a expression level? Based on the current data, CD107a expression did not match with GZMB and PRF. Any explanation for that? PRF and GZMB levels changed in the shRNA system, but K562 cells cannot be killed. Any explanation for the way that happened. How about other target cells or viral infection?

While the percentage of CD107a is significantly reduced when RUNX2 was silenced (Figure 5b), the expression level of CD107a (MFI) is not significantly altered, neither by knockdown nor by overexpression of RUNX2(-I). Please see Author response image 2 as information for the reviewer.

Cytoplasmic effector molecule expression in ‘resting’ NK cells and degranulation (cell membrane CD107a expression) upon target recognition are two different processes that are not necessarily similarly regulated. It is possible that NK cells increase expression of cytotoxic effector molecules, while degranulation is reduced. It is known that RUNX2 can act as a transcriptional activator as well as a repressor, depending on the location of the consensus sequence or the type of binding partners in the transcriptional complex. Our hypothesis is that RUNX2 activates expression of cytotoxic effector molecules but reduces degranulation.One possible explanation for the unaltered cytotoxic potential of RUNX2-silenced NK cells is that, while granzyme B and perforin expression are increased, degranulation is reduced, and these opposite effects might counterbalance each other. Although NK cells with RUNX2 knockdown did not exhibit changes in cytotoxicity towards K562 cells, it does not mean that the same is true for other target cells. However, we did not perform experiments with other target cells or viral infection.

5. Check the period in line 31. The color is different.

This is adjusted now (see track changes).

6. Check line 108; it should be "RUNX2 controls human NK cell development."

As indicated in our response to your first major comment, we were not able to demonstrate direct induction of IL-2Rβ by RUNX2, but our data show that RUNX2 affects IL-2Rβ expression in human NK cell development in vitro. We therefore suggest the following title of the paragraph: “RUNX2 controls human NK development possibly by regulating IL-2Rβ expression”

Significance:Defining the role of Runx2 in NK cell development and functions.Audience:Researchers working on transcription factors and NK cell biologists.Reviewer #2Summary:This is an interesting study that adds new information regarding a role for RUNX2 in NK cell development. Wahlen et al. present very interesting findings highlighting the role for RUNX2 in the acquisition of a tissueresident phenotype in differentiating NK cells. The authors demonstrate that RUNX2-I isoform is predominantly expressed in specific human NK cells subsets and that RUNX2 upregulates IL-2Rβ expression in NK cell-committed progenitors. Interesting results integrating ChIP-seq and RNAseq data and basic functional studies show that RUNX2 regulates several genes associated with NK cell tissue homing and recirculation. The authors postulate that RUNX2 regulates the acquisition of a tolerogenic tissue-resident phenotype in human NK cells. There are a number of intriguing observations that should be of interest in the field.Major comments:1. While the results obtained are interesting and scientifically sound, the manuscript does not rigorously prove that RUNX2 is involved in NK cell differentiation and development. The results were obtained using human cells ex vivo and in vitro human HSC-based cultures for NK cell differentiation and development. The authors would need a relevant model in vivo to fully characterize phenotypic and functional features of NK cells in the absence or presence of RUNX2. Such studies would be essential, in particular for the acquisition of a tissue-specific resident phenotype in human NK cells in distinct microenvironments. Furthermore, the authors should also modify extensively the title and several statements across the manuscript regarding the role for RUNX2 in NK cell differentiation and development.

Thank you for your very positive comments. We fully agree that an in vivo model is required to study the role of RUNX2 in differentiation of human NK cells and in their acquisition of a tissue specific resident phenotype in distinct organs. We therefore now performed an in vivo experiment in which we humanised lethally irradiated NSG-huIL-15 mice by intravenous injection of bulk CD34^+^ CB-derived HSC, which were transduced with either control or RUNX2 shRNA lentivirus. After 6-7 weeks, we analysed the presence of human eGFP^+^ NK cells (CD45^+^CD56^+^CD94^+^) as well as the frequency of tissue-resident (CD69^+^CD49e^-^) and circulating (CD69^-^CD49e^+^) human NK cells in the lungs, liver, spleen, bone marrow and lamina propria of the intestine (Figure 6A; Supplementary Figure 5). We found that the absolute number of human NK cells was drastically reduced in all organs of mice injected with RUNX2-silenced HPC compared to those injected with control HPC (Figure 6B). In addition, RUNX2 silencing significantly reduced the frequency of trNK cells in the bone marrow and lamina propria fraction, while it increased the percentage of circNK cells (Figure 6C). These data show that (1) also in vivo RUNX2 is an important transcription factor for human NK cell development and that (2) RUNX2 is involved in human NK cell tissue residency, at least in the bone marrow and in lamina propria of the intestine. With regard to the other examined organs, the frequency of tr- and circNK cells was unaffected by RUNX2 knockdown, except for the spleen where circNK cells were decreased. This suggests that either RUNX2-mediated regulation of NK cell tissue residency is tissue-specific (bone marrow and lamina propria) or that, at least for some organs, this mouse model is not representative for human biology. We have incorporated these new findings in the manuscript.

With the new insights that we gained after the additional experiments, we changed the title of the manuscript to ‘The transcription factor RUNX2 drives the generation of human NK cells and promotes tissue residency’. As the reviewer suggested, we also re-evaluated several statements across the manuscript regarding the role of RUNX2 in NK cell differentiation via regulation of IL2Rβ expression and in promoting tissue residency.

2. Additional evidence for the direct regulation of IL-2Rβ expression by RUNX2 would be helpful

We fully agree with your comment, which is why we tried to confirm the direct influence of RUNX2 on IL-2Rβ expression using an IL-2Rβ reporter assay. In this assay, we introduced both the IL-2Rβ reporter vector (Genecopoeia vector HPRM30810-LvPM03 with IL-2Rβ promoter sequence, 1585 bp in length, 269bp downstream of TSS), and the LZRS vector with RUNX2-I in a RUNX2^low^ IL-2Rβ^low^ ALL-SIL cell line. Instead of the RUNX2-I vector, control ALL-SIL cells received the empty LZRS vector together with the IL-2Rβ reporter construct. However, due to practical issues, we were unable to generate any conclusive results. Since we are unable to definitively demonstrate the direct induction of IL-2Rβ by RUNX2, we have nuanced our statements in the manuscript. Our data still show that RUNX2 affects IL-2Rβ expression in human NK cell development in vitro. We adjusted the text in the manuscript.

3. The studies showing that RUNX2 negatively regulates granzyme B, perforin expression and IFN-γ and TNF-α secretion are intriguing and could be better explored.

Although the role of Runx3 in the transcriptional regulation of granzyme B, perforin, IFN-γ and TNFα in murine CD8^+^ T and Th1 cells has been demonstrated, a similar role for Runx2 has not been described yet. For example, a study performed by Olesin *et al. (PMID 30264035)* showed that although Runx2 plays a role in the generation of murine CD8^+^ memory T cells, there was no impact on the expression of effector molecules and recall response. Furthermore, the regulation of these NK cell effector molecules by RUNX proteins in human NK cells remains unidentified. I agree that the underlying molecular mechanism of RUNX2-mediated regulation of effector molecule expression is certainly an interesting topic that should be thoroughly investigated. Since there are many mechanisms involved in regulation of the expression and secretion of effector molecules, it is almost a topic on its own and therefore part of a follow-up study.

Minor comments:1. "Taken together, we deduce from the data that RUNX2 promotes NK cell development by inducing IL-2Rβ expression and thereby enabling NK lineage commitment" – this is an overstatement

We fully agree with your comment. We have therefore adjusted our statement by concluding that RUNX2 promotes NK cell development in part by regulating IL-2Rβ expression and thereby promoting NK lineage commitment. You can find this adjustment in the discussion on page 11 line 273-274.

2. "It is well-known that RUNX2 by itself is a relatively weak transcription factor …." – this statement should be modified.

Our statement that RUNX2 is a weak transcription factor may indeed cause misconceptions. As RUNX2 by itself has a low affinity for DNA, it needs to form a complex with other co-factors such as CBFβ to increase the stability and affinity of the interaction with DNA. You can find the adjusted statement in the discussion (page 11 line 277-278).

Significance:To date, the role of RUNX2 in NK cell development has not been investigated in mice nor in humans. The findings will contribute to a better understanding of NK cell biology and may help in in improving existing NK cell-based therapies in the future. However, lack of relevant in vivo studies diminishes the importance of this work. Further studies are warranted.Audience:ImmunologistsExpertise in leukemia pathobiology and immunotherapies.Reviewer #3Summary:In this study, Wahlen et al. interrogated the role of RUNX2 in regulating human NK cell development through knockdown and overexpression studies. In agreement with previous work, the authors observed high RUNX2 expression in NK cell progenitors and a decline in expression levels in mature subsets. RUNX2 knockdown and overexpression was performed through viral transduction of cord blood-derived HPCs that were subsequently differentiated into NK cells in vitro. The authors found that RUNX2 knockdown led to a reduction in the numbers of mature NK cells, while overexpression had the opposite effect. They also provide data suggesting that RUNX2 may directly promote expression of the β chain of the IL-2 receptor during NK cell development. The authors also performed RNA-seq on sorted RUNX2 knockdown and overexpressing cells and compared this data to RNA-seq datasets that were generated using tissue-resident NK cells from liver and bone marrow. They identified Gene Set Enrichment signatures that were similar between tissue-resident NK cells and NK cells overexpressing RUNX2. Changes in the expression of several genes associated with circulation and residency were confirmed by flow cytometry. Finally, the authors performed function assays and found that manipulation of RUNX2 did not affect cytotoxicity, but overexpression reduced inflammatory cytokine production.Major comments:1. The title of the paper (The transcription factor RUNX2 promotes the development of human tissue resident NK cells) presents too strong of a conclusion that is not sufficiently supported by the data. While the NK cells differentiated in vitro with RUNX2 overexpression do appear to share a transcriptional signature with tissue-resident subsets and express receptors associated with tissue residency, the authors did not perform any adoptive transfer experiments showing that RUNX2 overexpressing NK cells are actually tissue resident. Such experiments would be necessary to support the conclusion stated in the title.

Indeed, since the original manuscript contained only data obtained from in vitro differentiation cultures, the concept of NK cell tissue residency required validation in an in vivo system. We therefore performed an in vivo experiment, in which we humanised lethally irradiated NSG-huIL15 mice by intravenous injection of bulk CD34^+^ CB-derived HSC, which were transduced with either control or RUNX2 shRNA lentivirus. After 6-7 weeks, we analysed the presence of human eGFP^+^ NK cells (CD45^+^CD56^+^CD94^+^) as well as the frequency of tissue-resident (CD69^+^CD49e^-^) and circulating (CD69^-^CD49e^+^) human NK cells in the lungs, liver, spleen, bone marrow and lamina propria of the intestine (Figure 6A; Supplemental Figure 5). We found that the absolute number of human NK cells was drastically reduced in all organs of mice injected with RUNX2-silenced HPC compared to those injected with control HPC (Figure 6B). In addition, RUNX2 silencing significantly reduced the frequency of trNK cells in the bone marrow and lamina propria fraction, while it increased the percentage of circNK cells (Figure 6C). These data show that (1) also in vivo RUNX2 is an important transcription factor for human NK cell development and that (2) RUNX2 is involved in human NK cell tissue residency, at least in the bone marrow and in lamina propria of the intestine. With regard to the other examined organs, the frequency of tr- and circNK cells was unaffected by RUNX2 knockdown, except for the spleen where circNK cells were decreased. This suggests that either RUNX2-mediated regulation of NK cell tissue residency is tissue-specific (bone marrow and lamina propria) or that, at least for some organs, this mouse model is not representative for human biology. We have incorporated these new findings in the manuscript.

With the new insights that we gained after the additional experiments, we changed the title of the manuscript to ‘The transcription factor RUNX2 drives the generation of human NK cells and promotes tissue residency’. As the reviewer suggested, we also re-evaluated several statements across the manuscript regarding the role of RUNX2 in NK cell differentiation via regulation of IL2Rβ expression and in promoting tissue residency.

2. The authors used RNA-seq data from tissue-resident NK cells in comparisons with their RUNX2 knockdown and overexpressing NK cells. Do they see elevated RUNX2 transcript levels in tissue resident NK cells? I don't know if matched circulating NK cell data is available, but such a finding would further strengthen the connection between the tissue residency profile and RUNX2.

Thank you for your very valid comment. We re-investigated the public RNA-seq data of tissue resident and circulatory NK cells of liver (Cuff *et al.*) and bone marrow (Melsen *et al.*). These data sets show that in the liver and bone marrow, RUNX2 transcript levels are indeed elevated in tissue resident NK cells compared to circulatory NK cells. The fold changes of RUNX2 in liver and bone marrow tissue-resident versus circulatory NK cells are 25 and 13, respectively. This supports our hypothesis and we have included this in the results on page 7 line 159-163

3. The evidence that RUNX2 controls human NK cell development by direct induction of IL-2Rbeta expression is fairly weak. In Figure 2a, it appears as though RUNX2 knockdown didn't significantly affect IL-2Rbeta expression, and RUNX2 overexpression only affected IL-2Rbeta expression at day 7 (not day 14). Furthermore, in Figure 2b, RUNX2 knockdown did not impact IL-2Rbeta expression in YTS cells. The authors speculate that residual RUNX2 may be sufficient to drive IL-2Rbeta expression. This could be tested by knocking out RUNX2 with a CRISPR-Cas9 system. The authors also provide some ChIP-seq data showing that RUNX2 binds to the promoter region of IL2RB in human PBNK cells but do not provide any ChIP data showing enhanced enrichment of RUNX2 within IL2RB in their RUNX2 overexpressing cells.

Indeed, RUNX2 knockdown did not result in significant changes of IL-2Rβ expression in NK cell progenitors or in the YTS cell line, which we attributed to residual RUNX2 expression. As suggested by the reviewer, we attempted to knockout RUNX2 in the YTS cell line using the CRISPR-Cas9 system (Synthego) to investigate the effect on IL-2Rβ expression. However, we were unsuccessful to obtain cells that lacked RUNX2 expression. So at this point we cannot state that RUNX2 is essential for IL-2Rβ expression in human NK cells or their progenitors.

The reviewer is also correct that the percentage of IL-2Rβ^+^ stage 3 cells upon RUNX overexpression in HSC was significantly increased on day 7, whereas this was no longer the case on day 14. However, this is not a counterargument for a regulating role of RUNX2 in IL-2Rβ expression as the subpopulation of stage 3 cells that expresses IL*-*2Rβ^+^ are NK cell-committed progenitors, that will probably have differentiated into stage 4 or stage 5 NK cells on day 14. This is in agreement with the increased absolute cell numbers of stage 4 and stage 5 NK cells in the RUNX2 overexpression cultures on day 14. We could not perform ChIP-seq on RUNX2 overexpressing stage 3 cells as this technique requires large cell numbers, which are not feasible to generate in vitro.

Thus, although we do not state that RUNX2 is essential for IL*-*2Rβ expression, our data from the RUNX2-I overexpression model and RUNX2-specific ChIP-seq do provide evidence for a certain degree of RUNX2-mediated regulation of IL*-*2Rβ expression during human NK cell development. We therefore downgraded our statements regarding this matter in the manuscript.

Minor comments:No minor commentsSignificance:As an NK cell biologist suggest returning for major revision and re-evaluation.